



# A comparative study of fabric evolution models and anisotropic rheologies

Daniel H. Richards[1,2], Elisa Mantelli[3,4,1], Samuel S. Pegler[5], and Sandra Piazolo[6]

[1]The Australian Centre for Excellence in Antarctic Science, University of Tasmania, Hobart, Australia
[2]School of Earth, Atmosphere and Environment, Monash University, Melbourne, Australia
[3]Department of Earth and Environmental Sciences, Ludwig-Maximillians-Universitaet, Munich, Germany
[4]Glaciology Section, Alfred Wegener Institute Helmholtz Centre for Polar and Marine Research, Bremerhaven, Germany
[5]School of Mathematics, University of Leeds, United Kingdom
[6]School of Earth and Environment, University of Leeds, United Kingdom

**Correspondence:** Daniel H. Richards (daniel.richards@utas.edu.au)

**Abstract.**

Ice is anisotropic, with its viscosity varying by an order of magnitude in different directions when ice crystals align. However, how this variation affects ice flow is not well understood. This is because of a lack of a) models for fabric (the collective distribution of crystal orientations) evolution accurate enough to reproduce observations, and b) knowledge of which anisotropic rheology is most appropriate. Here we address both these problems. First, we review a range of previous models for fabric evolution and show they can be combined into a common differential equation. This incorporates a handful of parameters and an anisotropic rheology, which can be freely chosen. We apply this model, with a range of different anisotropic rheologies, to both an ice stream and an ice divide. For each rheology we choose the parameters to give the best possible fit to observations. We find these parameters are significantly different from those used previously. Best results come from assuming the grains rotate due to stress rather than deformation, with the stress calculated through an anisotropic rheology. By including grain rotation primarily due to stress, combined with a diffusion of the fabric, we can reproduce observations at both an ice divide and, for the first time, at an ice stream. We also compare and rank a range of anisotropic rheologies based on the accuracy of their fabric predictions. The rheologies which give the closest fit to observations have a tensor description of the anisotropy and assume that neighbouring ice grains experience approximately the the same stress.

## 1 Introduction

The conventional approach to modelling the flow of glaciers and ice sheets is to consider ice as a continuum, viscous fluid obeying the Stokes equations (or some simplification thereof). However, at a small enough scale ice is not a traditional fluid, but rather a polycrystalline solid constituted of grains with a regular, hexagonal, crystallographic structure. At the grain scale, ice deforms in a very different way to liquids: while liquids deform by the free movement of molecules past one another, ice deforms through the movement of line defects along particular planes in the crystal lattice. Unsurprisingly, this deformation occurs over much longer timescales. Because of the hexagonal lattice and molecular structure of natural ice, it is around 70





times easier for these defects to move within a particular plane - along the *basal plane* (Duval et al., 1983). Consequently, the collective orientation of these basal planes - the *fabric* - can cause the mechanical properties of ice to vary by an order of magnitude in different directions (e.g. Pimienta and Duval, 1987). This directional variation is called anisotropy.

Physically, the effect of anisotropy on ice flow can be conceptualised as follows: for a given ice velocity field, the fabric will evolve as grains rotate due to the movement of dislocations within them, under a stress field . The new fabric then feeds back into the ice flow problem by altering locally the bulk mechanical properties of the ice. Mathematically, these physics are described by (i) an evolution equation for the fabric as a function of a given velocity field, and (ii) a constitutive relationship between stress and deformation (that is, a *rheology* or *rheology*) that depends on the fabric through a suitably defined viscosity.

Despite the order of magnitude effect on the ease of flow, and the fact that fabrics affect flow everywhere, the number of studies including this coupled fabric-flow evolution can be counted on one hand and has been limited to idealised flow line models, either at ice divides (Martín et al., 2009; Martín and Gudmundsson, 2012), simplified ice sheets (Ma et al., 2010) or ice streams (Lilien et al., 2021). When considering real-world domains, the effect of fabric is usually only represented by adding to the viscosity a constant enhancement factor which softens the ice everywhere by around a factor of 8. This, rather than a

coupled fabric-flow approach, is the approach taken by almost all large-scale ice sheet simulation codes whenever they account for anisotropic effects (e.g Winkelmann et al., 2011; Larour et al., 2012).

    A key difficulty in modelling fabrics and their effect arises from our lack of understanding of the underlying physics. Historically, progress has been limited by a lack of real-world observational data with which to compare models to. Despite recent advances in radar and seismics for measuring fabrics from the surface or air (e.g. Smith et al., 2017; Jordan et al., 2020;

Kufner et al., 2023), ice cores still provide by far the most complete dataset. Unfortunately, ice cores have traditionally been drilled at ice divides (for a review see Faria et al., 2014a, and refs. therein), locations specifically chosen for their minimal and simple deformation in order to give accurate paleo-climate records. While we now have lots of data on the fabrics produced at ice divides, information is lacking for all other areas of the ice sheet. In order to fill this gap, Laboratory experiments have been carried out measuring fabrics produced under applied deformations (e.g. Budd et al., 2013; Craw et al., 2018; Journaux

et al., 2019, and references therein). Yet experiments need to take place at stresses orders of magnitude greater than those seen in the natural world to allow for experiments that last for reasonable durations (days to weeks), limiting their applicability to natural ice flow. However recently, the EGRIP project (Stoll et al., 2021) has drilled an ice core in the much more dynamic environment of an ice stream. This represents the first complete fabric dataset for which the observed fabric is a product of a complex and changing deformation regime. Consequently, we now have full fabric data in a range of locations and deformation

regimes in the natural world, which opens up new possibilities for developing and testing coupled anisotropy models.

    Currently, there are two main problems in modelling anisotropy. Firstly, confidence in the ability of existing models for fabric evolution to accurately represent fabric for complex stress regimes remains limited. While models have been able to accurately predict fabric evolution at ice divides (Montagnat et al., 2014) and in experiments (Richards et al., 2021), recent work showed difficulties predicting fabrics in the more dynamic regions of ice sheets mentioned above (Richards et al., 2023),

suggesting better physical assumptions are required. Secondly, there has been no way to test whether an anisotropic rheology is accurately representing the anisotropic effect. This lack of testing has led to a proliferation of competing rheologies (Gillet-





Chaulet et al., 2005; Placidi et al., 2010; Graham et al., 2018; Rathmann and Lilien, 2022), with no consensus on which may be most appropriate.

In this contribution, we aim to address these problems by providing both accurate modelling of ice fabrics in a range of natural conditions, and a test of the accuracy of competing rheologies. To do this, we first review and analyse the underlying physics in existing models of anisotropy, through the dynamics of an individual grain and the link between the grain scale and ice sheet scale. Physically, it is known that grains in ice are closer to the limit of experiencing uniform stress rather than uniform deformation (Castelnau et al., 1996a), and we trace this knowledge through to the large-scale models.

    We develop a unified framework for fabric evolution, which can express a range of previous models of anisotropy as cases
of a common equation (sections 2, 3, 4). This equation depends on a few parameters, and the anisotropic rheology used. This allows us to compare different rheologies against each other, with the hypothesis that the most realistic rheology provides the most accurate fabric predictions. We apply the fabric predictions from these rheologies to two locations: an ice divide and an ice stream (sections 5 and 6), representing very different deformation conditions in an ice sheet.

## 2   Grain dynamics

The movement of glaciers and ice sheets, over centuries and millennia, is fundamentally governed by the movement of dislocations through grains (regions of similar lattice orientation) (Duval, 1981). Dislocations are line defects in the crystal lattice shape caused by impurities or a local excess or deficiency of hydrogen atoms (Glen, 1968). Under an applied stress, dislocations are generated and move in a certain direction along certain slip planes within the crystal lattice. Therefore, the deformation and thus movement of glaciers and ice sheets at the largest scales is governed by the dynamics of dislocations deforming individual
grains. For ice, slip along the basal plane - with unit normal denoted by the $c$-axis - is approximately 70 times easier than other slip systems (Duval et al., 1983).

    We examine a range of models that all treat the grain as the smallest physical element, i.e. all quantities are uniform within the grain. A grain can be described by its mass $m$, its position $\boldsymbol{x}$ and its orientation. Due to the ease of dislocation glide along the basal plane, all the models we examine make the assumption that the basal plane is the only plane dislocations glide along.
Consequently, the grains orientation can be described solely by the basal plane normal, the $c$-axis, $\boldsymbol{c}$, and other crystallographic axes can be neglected. The direction of a $c$-axis can be described by 2 angles or, equivalently, by a unit vector in Cartesian space, and the space of possible $c$-axis orientations is described by the unit sphere.

### 2.1   Grain rheology

A constitutive law for the grain gives the deformation $\dot{\boldsymbol{\varepsilon}}'$ of the grain under an applied stress $\boldsymbol{\tau}'$, or vice versa, where dashes
denote grain quantities. The most complete approach to the grain rheology is to consider the contribution of all possible slip systems (e.g. Schmid and Casey, 1986). However, the models we discuss consider only dislocation movement along the basal plane. Therefore, with this assumption the most general model for grain rheology is to consider the grain as a transversely isotropic medium, which was first developed for metals by Johnson (1977) and first applied to ice by Meyssonnier and Philip



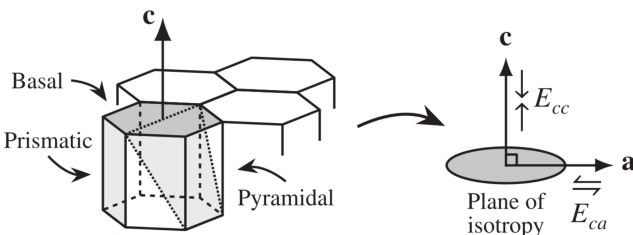

**Figure 1.** We model grains as transversely isotropic, shown on the right. This means we negelct over slip systems shown on the left, and consider only the basal plane, described by its normal $c$. The grain then has a 'plane of isotropy' within the basal plane, where the response is the same, and the overall response can be described by an enhancement in compression $E_{cc} \approx 1$ and and enhancement in shear $E_{ca} \gg 1$.

(1996). Using the notation of Rathmann et al. (2021) the grain is described by two 'enhancement factors' (Figure 1): An enhancement factor for deformation in the basal plane, $E_{ca}$, and an enhancement factor for deformation normal to the basal plane, $E_{cc}$:

$$\dot{\varepsilon}'_{cc} = \eta^{-1} E_{cc} \tau'_{cc}, \quad \dot{\varepsilon}'_{ca} = \eta^{-1} E_{ca} \tau'_{ca} \tag{1}$$

where $c$ is the direction of the $c$-axis, and $a$ is any direction perpendicular to this (Figure 1). While for an isotropic material, $E_{cc}$ and $E_{ca}$ would both be unity, for ice $E_{cc} \approx 1$, and $E_{ca} \gg 1$, reflecting the ease of deformation along the basal plane. For a linear grain behaviour, the rheology is:

$$\dot{\varepsilon}' = \eta^{-1} \left( \boldsymbol{\tau}' + \frac{3(E_{cc}-1) - 4(E_{ca}-1)}{2}(\boldsymbol{\tau}':\boldsymbol{cc})\boldsymbol{cc} + (E_{ca}-1)(\boldsymbol{\tau}' \cdot \boldsymbol{cc} + \boldsymbol{cc} \cdot \boldsymbol{\tau}') - \frac{E_{cc}-1}{2}(\boldsymbol{\tau}':\boldsymbol{cc})\mathbf{I} \right) \tag{2}$$

where, for this case of linear grains $\eta^{-1} = A$, and $\mathbf{I}$ is the 2nd-rank identity tensor. This rheology has been in Thorsteinsson (2002); Gödert (2003), and simplifies to the grain rheology used in (Azuma and Goto-Azuma, 1996) if the further assumption of keeping only terms of $O(E_{ca})$ is made. It has been extended to the non-linear case in (Rathmann et al., 2021), however this introduces terms that make it computationally infeasible to include in a large-scale model. Consequently, all large-scale models have so far been limited to linear grains, and we do the same in this work. This does not preclude the use of power-law rheology above the grain scale.

## 2.2 Grain rotation

Equation (2) gives a constitutive law for grain behaviour linking the deformation $\dot{\boldsymbol{\varepsilon}}'$ and the stress $\boldsymbol{\tau}'$. We must now consider how the grains orientation $c$ changes under an applied deformation. In reality, a grain will always be embedded inside a wider network of grains. We consider a grain experiencing a known deformation $\dot{\boldsymbol{\varepsilon}}'$, as well as a known rotation, described by a rotation tensor $\boldsymbol{\omega}'$, describing the grain's rotation relative to a fixed reference frame.





The total change in rotation of a grain is a combination of the bulk rotation tensor and the plastic rotation tensor $\boldsymbol{\omega}'_p$:

$$\frac{d\boldsymbol{c}}{dt} = (\boldsymbol{\omega}' + \boldsymbol{\omega}'_p) \cdot \boldsymbol{c} \tag{3}$$

This plastic rotation is illustrated for an extensional deformation in Figure 2. The diagonal lines represent basal planes, along which the ice deforms 'like a deck of cards' (McConnel, 1891). This induces the plastic rotation $\boldsymbol{\omega}'_p$. Dafalias (1984) gives the mathematical form for $\boldsymbol{\omega}'_p$. It is also assumed that the basal planes remain parallel under a simple shear deformation (Gödert and Hutter, 1998):

$$\boldsymbol{\omega}'_p = \boldsymbol{cc} \cdot \dot{\boldsymbol{\varepsilon}}' - \dot{\boldsymbol{\varepsilon}}' \cdot \boldsymbol{cc} \tag{4}$$

combining Eqs. (3) and (4) gives:

$$\frac{d\boldsymbol{c}}{dt} = \boldsymbol{\omega}' \cdot \boldsymbol{c} - (\dot{\boldsymbol{\varepsilon}}' \cdot \boldsymbol{c} - (\dot{\boldsymbol{\varepsilon}}' : \boldsymbol{cc})\boldsymbol{c}), \tag{5}$$

If a model of transversely isotropic grains is used then the expression for grain rheology (Eq. (2)) can be substituted into Eq. (5) to express the lattice rotation in terms of grain stress and grain enhancements:

$$\frac{d\boldsymbol{c}}{dt} = \boldsymbol{\omega}' \cdot \boldsymbol{c} - E_{ca}\eta^{-1}(\boldsymbol{\tau}' \cdot \boldsymbol{c} - (\boldsymbol{\tau}' : \boldsymbol{cc})\boldsymbol{c}), \tag{6}$$

## 2.3 Recrystallization

### 2.3.1 Migration recrystallization

A grain is described by its orientation $\boldsymbol{c}$ and mass $m$ and constitutive relationship linking deformation and stress. We have reviewed the common equations used in to describe the constitutive law and change in orientation of a grain. We now consider how the mass of a grain $m$ evolves. While grains undergoing no deformation may undergo normal grain growth, in an ice sheet

which is always deforming, the primary process for grain growth will be migration recrystallization (Faria et al., 2014a).

Placidi et al. (2010) considered the effect of migration recrystallization on a continuum description of the fabric. In this sub-section, we aim to more explicitly explore how the equation of Placidi et al. (2010) can be derived by considering the dynamics of an individual grain.

Migration recrystallization is the process where one grain grows at the expense of another. Hence, whereas previously we

considered a single grain we must now consider a system of $N$ grains. Migration recrystallization is driven by the minimisation of potential energy of the system. Grains with a higher dislocation density have higher potential energy. Consequently, the change in mass of a grain $\dot{m}$ is a function of this dislocation density $\rho_D$ and the grains mass $m$:

$$\dot{m} = f(\rho_D, m), \quad \frac{\partial f}{\partial \rho_D} < 0. \tag{7}$$

The total mass of the system must also be conserved:

$$\sum_{i=1}^{N} \dot{m}_i = \sum_{i=1}^{N} f(\rho_{Di}, m_i) = 0 \tag{8}$$





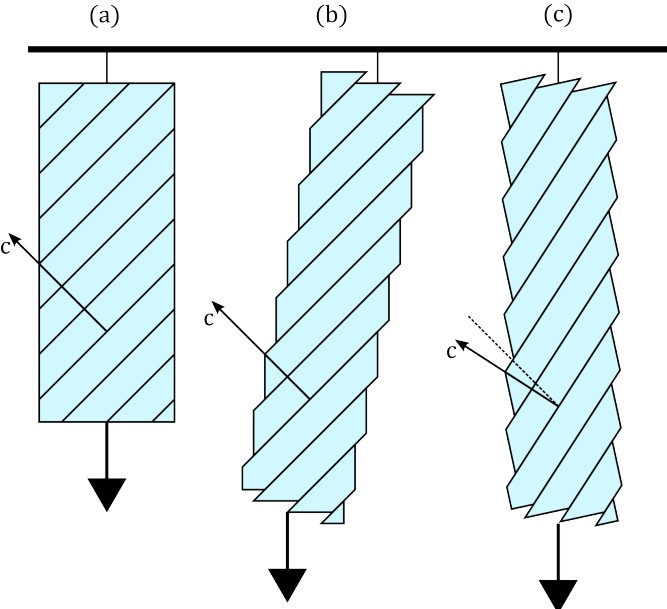

**Figure 2.** Illustration of an ice grain deforming in tension, adapted from Placidi (2005). (a) the initial undeformed grain, with diagonal lines representing basal planes, and $c$-axis shown. (b) slide alone the basal planes deforms the grain under tension, however here the crystal is not in equilibrium as there is a couple acting on it. (c) the equilibrium position involves grain rotation as well, giving a change in the $c$-axis orientation.

Dislocations exist at the molecular scale so cannot be modelled directly. The dislocation density, and hence migration recrystallization, is also likely related to the history of the grain. However, the state-of-the-art for modelling migration recrystallization at the large scale assumes it is a function of the instantaneous properties (Placidi et al., 2010). To arrive at this result we introduce a proxy for dislocation density $\rho_D$ such that is a function of the grain properties described previously (i.e. the stress $\boldsymbol{\tau}'$, deformation $\dot{\boldsymbol{\varepsilon}}'$, mass $m$ and orientation $c_i$). Humphreys and Hatherly (2004) propose that the dislocation density is related to the direction of the stress acting on the grain, i.e. $\rho_D = g(\boldsymbol{\tau}', \boldsymbol{c})$. For example, if the stress imposed on the grain is in the direction of the basal plane, the dislocations can move easily to deform the grain. However, if the stress imposed on the grain is perpendicular to the basal plane, any existing dislocations cannot move easily. Consequently, new dislocations will nucleate to accommodate the applied stress, increasing the dislocation density. We can express this mathematically as:

$$\rho_D = g(s'), \quad \frac{\partial g}{\partial s'} < 0 \tag{9}$$

where $s'$ is the resolved stress on the basal plane, and the dislocation density decreases as the resolved stress increases (i.e. the stress is in the direction of the basal planes). The resolved stress can be expressed as (Placidi et al., 2010):

$$s' = \frac{(\boldsymbol{\tau}' \cdot \boldsymbol{c}) \cdot (\boldsymbol{\tau}' \cdot \boldsymbol{c}) - (\boldsymbol{\tau}' : \boldsymbol{cc})^2}{\boldsymbol{\tau}' : \boldsymbol{\tau}'}, \tag{10}$$





Combining Eqs. (7) and (9) gives:

$$\dot{m} = h(s', m), \quad \frac{\partial h}{\partial s'} > 0 \tag{11}$$

Based on Placidi et al. (2010), we then make the simplest possible assumption, that the function $h(s', m)$ can be approximated by multiplying the two variables together:

$$h(s', m) = \beta m s' - C \tag{12}$$

where $\beta$ is a constant of proportionality and $C$ can be found through conservation of mass, Eq. (11) becomes:

$$\dot{m}_i = \beta m_i \left( s'_i - \frac{\sum_{i=1}^{N}(m_i s'_i)}{\sum_{i=1}^{N} m_i} \right) \tag{13}$$

Where the subscript $i$ denotes an individual grain. To simplify this equation, we can replace $m$ with a mass fraction $w$:

$$w_i = \frac{m_i}{\sum_{i=1}^{N} m_i} \tag{14}$$

to finally give an equation instead for the change in mass fraction $dw/dt$:

$$\frac{dw}{dt} = \beta w(s' - \langle s' \rangle) \tag{15}$$

where $\langle s' \rangle$ denotes a weighted average over $N$ grains $\sum_{i=1}^{N} w_i s'_i$. As we show in Section 3, this is equivalent to the term derived by Placidi et al. (2010) for migration recrystallization, with the exception that $s'$ in Placidi et al. (2010) is a function of the large-scale deformation. The parameter $\beta$ has been shown to be strongly dependent on temperature (Richards et al., 2021) and, to a lesser extent, strain rate.

### 2.3.2 Rotational recrystallization

We are reviewing models which assume a grain has a uniform stress $\boldsymbol{\tau}'$ and deformation $\dot{\varepsilon}'$. However, this is a simplification. In reality, it has been observed that as a grain is deformed it tends to have a less stressed inner 'core' and a more stressed outer 'mantle' (Faria et al., 2009). Under stress, subgrains - small changes in orientation within the grain - form and grow in this outer region as a result of increased dislocation density and recovery (Drury and Urai, 1990). As strain increases, eventually a subgrain will have a sufficiently different ($\sim 10°$) orientation from the core grain that it can be called a new grain. This process

is called rotational recrystallization. The subgrains orientation will depend on the orientation of its parent grain, the stress on the grain and the orientation of its neighbours (e.g. Piazolo et al., 2015). At the large scale, this acts to diffuse concentrations in the fabric.

However the models we are reviewing, which aim to be included in large-scale ice sheet models, all neglect heterogenities within a grain as well as the interactions of neighbouring grains for reasons of computational efficiency. Gödert (2003) first

proposed incorporating the effect of rotational recrystallization by incorporating an additional Brownian motion term alongside the lattice rotation term $d\boldsymbol{c}/dt$ on the change in grain orientation away from the parent grain:

$$\frac{d\boldsymbol{c}}{dt} = \boldsymbol{v} + \sqrt{2\lambda}\mathbf{W}_Q, \tag{16}$$





where $\boldsymbol{v}$ represents the contribution to $d\boldsymbol{c}/dt$ from basal-slip deformation such as the right hand side in Eq. (6), $\lambda$ is a parameter and $\mathbf{W}_Q(t)$ is a multivariate Wiener process describing Brownian motion. This can be thought of as subgrains causing the average orientation of the grain as a whole to change in a 'random' way. This Browninan motion corresponds, as in molecular dynamics, to a diffusion at a larger scale - in this case of fabric concentrations. While the change in orientation would not be random if grain interactions could be modelled, observations from ice cores Durand et al. (2008) suggest it appears random when compared to the large scale deformation.

## 3   Moving from a grain to an ice sheet

So far, we have reviewed models for an isolated grain, internally uniform and subjected to externally imposed stresses or deformations. We collected equations for the rheology (Meyssonnier and Philip, 1996; Rathmann et al., 2021), change in $c$-axis orientation due to deformation (Azuma and Goto-Azuma, 1996) or rotational recrystallization (Gödert, 2003), and change in grain mass (Placidi et al., 2010) into a common model for an ice grain:

$$\dot{\boldsymbol{\varepsilon}}' = \eta^{-1}\left(\boldsymbol{\tau}' + \frac{3(E_{cc}-1)-4(E_{ca}-1)}{2}(\boldsymbol{\tau}':\boldsymbol{cc})\boldsymbol{cc} + (E_{ca}-1)(\boldsymbol{\tau}'\cdot\boldsymbol{cc}+\boldsymbol{cc}\cdot\boldsymbol{\tau}') - \frac{E_{cc}-1}{2}(\boldsymbol{\tau}':\boldsymbol{cc})\mathbf{I}\right) \tag{17}$$

$$\frac{d\boldsymbol{c}}{dt} = \boldsymbol{v} + \sqrt{2\lambda}\mathbf{W}_Q, \quad \boldsymbol{v} = \boldsymbol{\omega}'\cdot\boldsymbol{c} - (\dot{\boldsymbol{\varepsilon}}'\cdot\boldsymbol{c} - (\dot{\boldsymbol{\varepsilon}}':\boldsymbol{cc})\boldsymbol{c}) = \boldsymbol{\omega}'\cdot\boldsymbol{c} - E_{ca}\eta^{-1}(\boldsymbol{\tau}'\cdot\boldsymbol{c} - (\boldsymbol{\tau}':\boldsymbol{cc})\boldsymbol{c}) \tag{18}$$

$$\frac{dw}{dt} = \beta w(s' - \langle s'\rangle), \quad s' = \frac{(\boldsymbol{\tau}'\cdot\boldsymbol{c})\cdot(\boldsymbol{\tau}'\cdot\boldsymbol{c}) - (\boldsymbol{\tau}':\boldsymbol{cc})^2}{\boldsymbol{\tau}':\boldsymbol{\tau}'} \tag{19}$$

The above equations could be solved by placing the grains in a network in space, and deriving more equations for their interactions in a similar fashion to micro-mechanical models (e.g. Castelnau et al., 1996a; Piazolo et al., 2010).

However, as discussed previously, if we are concerned with an ice sheet or glacier, this is numerically infeasible as there are around $10^{12}$ grains in each cubic kilometre. Instead, what is done is to make a continuum assumption, that at each point in the material there are a large number of grains. This is analogous to considering a glass of water as a continuum fluid rather than a collection of water molecules. Like the velocity of a fluid at a point is the mean velocity of all the molecules, so small that they can be considered all 'at that point', in our case most quantities can be found by taking a weighted average of all the grains at that point, e.g. for the deformation tensor:

$$\dot{\boldsymbol{\varepsilon}} = \langle\dot{\boldsymbol{\varepsilon}}'\rangle = \sum_{i=1}^{N} w_i\dot{\boldsymbol{\varepsilon}}_i \tag{20}$$

However, continuing the analogy with a glass of water, when this continuum approximation is made emergent quantities occur: for example, the temperature of an individual water molecule is not a relevant concept but the temperature of the glass of water as a whole is a key control of its physics. The emergent quantity in this case is the *fabric*. The fabric is the





distribution of grain orientations in a region. It can be represented by a probability distribution function, often called the orientation distribution function $f$. This is different from a usual distribution function in that it is defined over the space of possible orientations, i.e. over two angles prescribing the surface of a sphere. In the real world, fabrics are measured when ice cores are drilled and the orientation of each grain within the core is measured. These can be considered as samples from the distribution function. If the dynamics of one sample are described by Eqs. (18) and (19), the Feynman-Kac formula (Kac, 1949) gives the corresponding partial differential equation for the probability distribution function $f$:

$$\frac{df(\boldsymbol{n})}{dt} = -\nabla^* \cdot (f\boldsymbol{v}) + \lambda \nabla^{*2}(f) + \beta f(s' - \langle s' \rangle), \tag{21}$$

where $\boldsymbol{v}$ and $s'$ are functions over the continuous angle $\boldsymbol{n}$ rather than discrete angle $\boldsymbol{c}$ and $\nabla^*$ is the gradient operator over $\boldsymbol{n}$ (not physical space $\boldsymbol{x}$) restricted to the space of possible orientations:

$$\nabla^* = \nabla - (\nabla \cdot \boldsymbol{n})\boldsymbol{n}. \tag{22}$$

This equation has been used before a number of times, with differing forms for $\boldsymbol{v}$, and sometimes without migration recrystallization (Gödert, 2003; Faria, 2006; Placidi et al., 2010; Richards et al., 2021; Rathmann et al., 2021).

The problem we have with this equation is that $\boldsymbol{v}$ and $s'$ depend on grain quantities. However when viscous anisotropy - controlled by fabric - is included in ice sheet models, it is included by coupling the equations for anisotropy to a large scale flow. At each timestep, the input to the anisotropic model is the flow field, from which the deformation tensor $\dot{\varepsilon}$ can be trivially calculated. The output from the model of anisotropy will be some viscosity, described by a fourth-rank tensor, such that the equation for the flow can be solved for the next timestep.

If we only have have macroscopic quantities as inputs, we need to recast Eqs. (17) - (19) in terms of these these macroscopic deformations and stresses rather than grain quantities. Equation (21) can then expressed in macroscopic quantities only.

## 3.1 The Taylor and Sachs approximations

To express Eq. (18) and Eq. (19), and hence Eq. (21) in macroscopic quantities, there are two possible assumptions available: either all grains deform at the same rate $\dot{\varepsilon}' = \langle \dot{\varepsilon}' \rangle = \dot{\varepsilon}$, or all grains experience the same stress $\boldsymbol{\tau}' = \langle \boldsymbol{\tau}' \rangle = \boldsymbol{\tau}$ . These two approximations, which acts as two bounds the true grain behaviour will lie between, are called the *Taylor* and *Sachs* bounds respectively. Polycrystalline materials which deform close to the Taylor bound have many easy slip systems, such as some metals (e.g Thamburaja and Jamshidian, 2014). This means that dislocations can move in different directions, and the applied deformation can be easily accommodated by the grains. However, ice has only 1 easy slip system: the basal plane. When applied to ice, the Taylor approximation predicts around 60% of the strain is due to non basal slip, and weak fabrics (Castelnau et al., 1996b, a). Observations from ice sheets which predominantly show strong fabrics, and the fact that slip along the basal plane is around 70 times easier than other slip systems (Duval et al., 1983) provide evidence that ice is deforming closer to the Sachs bound than the Taylor.





### 3.1.1 Sachs model

Consequently, the Sachs approximation $\boldsymbol{\tau}' = \langle \boldsymbol{\tau}' \rangle = \boldsymbol{\tau}$ more closely represents reality for ice, and is used by Azuma and Goto-Azuma (1996) and (Gödert, 2003) to derive a macroscopic rheology. The Sachs rheology can be derived by applying the averaging operater $\langle \cdot \rangle$ to Eq. (17):

$$\dot{\boldsymbol{\varepsilon}} = \eta^{-1} \left( \boldsymbol{\tau} + \frac{3(E_{cc}-1) - 4(E_{ca}-1)}{2} \boldsymbol{\mathcal{A}} : \boldsymbol{\tau} + (E_{ca}-1)(\boldsymbol{\tau} \cdot \mathbf{A} + \mathbf{A} \cdot \boldsymbol{\tau}) - \frac{E_{cc}-1}{2} \mathbf{I} \mathbf{A} : \boldsymbol{\tau} \right) \tag{23}$$

where $\mathbf{A}$ is defined below in Eq (26). For linear grain behaviour, $\eta^{-1} = A$. However Eq. (23) has been extended to include a power-law response by Pettit et al. (2007) by setting:

$$\eta^{-1} = A(\boldsymbol{\tau} : \boldsymbol{\tau})^{(n-1)/2} \tag{24}$$

and by Martín et al. (2009) by setting:

$$\eta = A^{-1/n}(\dot{\boldsymbol{\varepsilon}} : \dot{\boldsymbol{\varepsilon}})^{(1-n)/2n}, \tag{25}$$

allowing a macroscopic power law response through $n$ even if we assume the grains behave with a linear response. In Eq. (23) $\mathbf{A}$ is the second-order orientation tensor:

$$\mathbf{A} = \langle \boldsymbol{cc} \rangle = \sum_{i=1}^{N} w_i \boldsymbol{c}_i \boldsymbol{c}_i = \int_{S^2} f(\boldsymbol{n}) \boldsymbol{nn} \, d\boldsymbol{n} \tag{26}$$

As can be seen from this equation, it is also the second moment of the orientation distribution function $f$, and is a second-rank tensor. The eigenvalues of $\mathbf{A}$ are an important measure of the fabric. Similarly, the fourth-order orientation tensor is:

$$\boldsymbol{\mathcal{A}} = \langle \boldsymbol{cccc} \rangle = \sum_{i=1}^{N} w_i \boldsymbol{c}_i \boldsymbol{c}_i \boldsymbol{c}_i \boldsymbol{c}_i = \int_{S^2} f(\boldsymbol{n}) \boldsymbol{nnnn} \, d\boldsymbol{n} \tag{27}$$

Note for the case $E_{cc} = 1$ and $E_{ca} = 1$, Eq. (23) simplifies to Glen's rheology.

While the Sachs rheology has been used before, in this contribution we use it in fabric predictions and compare it to other rheologies. To do this, its helpful to introduce a normalised deviatoric stress $\hat{\boldsymbol{\tau}}$, such that subject to some normalisation factor for each rheology, for an isotropic fabric and only for an isotropic fabric (this means the grains making up the fabric have random orientations):

$$\hat{\boldsymbol{\tau}} = \dot{\boldsymbol{\varepsilon}}. \tag{28}$$

This can be found by substituting orientation tensors corresponding to an isotropic fabric ($A_{ij} = \delta_{ij}/3$ and $A_{ijkl} = (\delta_{ij}\delta_{kl} + \delta_{ik}\delta_{jl} + \delta_{il}\delta_{jk})/15$) into Eq. (23), giving:

$$\dot{\boldsymbol{\varepsilon}} = \frac{1}{0.4E_{ca} + 0.2E_{cc} + 0.4} \left( \hat{\boldsymbol{\tau}} + \frac{3(E_{cc}-1) - 4(E_{ca}-1)}{2} \boldsymbol{\mathcal{A}} : \hat{\boldsymbol{\tau}} + (E_{ca}-1)(\hat{\boldsymbol{\tau}} \cdot \mathbf{A} + \mathbf{A} \cdot \hat{\boldsymbol{\tau}}) - \frac{E_{cc}-1}{2} \mathbf{I} \mathbf{A} : \hat{\boldsymbol{\tau}} \right) \tag{29}$$





This is a linear equation and if $\dot{\varepsilon}$ is known, it can be inverted easily to find $\hat{\tau}$. Note that Eq. (29) does not depend on $\eta^{-1}$, and hence it does not depend on $n$.

Equations. (18) and (19) can also be expressed in macroscopic quantites using the Sachs approximation. It is also necessary to approximate the grain rotation $\omega'$. For this, the approximation $\omega' = \omega = (\nabla u - \nabla u^T)/2$, i.e. the grain rotation is equal
to the bulk rotation, is made as no other information is available. Eq. (18) can be combined with Eq. (6) to express $v$ in macroscopic quantities:

$$v = \omega \cdot c - \iota(\hat{\tau} \cdot c - (\hat{\tau} : cc)c) \tag{30}$$

where

$$\iota = \frac{E_{ca}}{0.4E_{ca} + 0.2E_{cc} + 0.4}. \tag{31}$$

Note that Eq. (30) now depends asymptotically on $E_{ca}$, as $E_{ca} \to \infty$ $\iota \to 2.5$ (for $E_{cc} = 1$).

Similarly the expression for $s'$ in Eq. (19) can be expressed in macroscopic quantities:

$$s' = \frac{(\hat{\tau} \cdot c) \cdot (\hat{\tau} \cdot c) - (\hat{\tau} : cc)^2}{\hat{\tau} : \hat{\tau}} \tag{32}$$

Because both Eq. (30) and (32) can be written solely depending on $\hat{\tau}$ rather than $\tau$, the fabric evolution predictions do not depend on the value of $n$.

### 3.1.2   Taylor model

If the Taylor approximation is used ($\dot{\varepsilon}' = \langle \dot{\varepsilon}' \rangle = \dot{\varepsilon}$) then $v$ from Eq. (18) can be expressed in macroscopic quantities:

$$v = \omega \cdot c - (\dot{\varepsilon} \cdot c - (\dot{\varepsilon} : cc)c) \tag{33}$$

A macroscopic equation for $s'$ is more complicated with the Taylor approximation and requires a constitutive law. However, as we test the model in only low temperature environments, where migration recrystallization is insignificant such that we set
$\beta = 0$ (see Section 6.1) we have not derived this term.

## 4   Macroscopic models

In the previous section we have reviewed two models for anisotropy based on assumptions about grain dynamics: the Taylor and Sachs model. In particular, we have shown that for the Sachs model the fabric evolution equation Eq. (30) depends on the stress, which must be found through a rheology if we start with the velocity hence deformation $\dot{\varepsilon}$.

We aim to test the set anisotropic rheologies aiming to be included in ice sheet models, through their effect on fabric predictions. Consequently, we now review these other models. Instead of considering the dynamics of grains directly, these models consider equations directly at the macroscopic scale.



## 4.1 Orthotropic rheologies

Gillet-Chaulet et al. (2005) and Rathmann and Lilien (2022) derive anisotropic rheologies not by assuming an ice grain has a
certain stress or deformation, but by assuming that the rheology has certain symmetries, specifically that it is invariant under
reflections along symmetry directions of the fabric. As these rheologies work form assumptions about the fabric rather than the
grain, they can be called macroscopic.

### 4.1.1 General Orthotropic Linear flow law (GOLF)

The GOLF rheology (Gillet-Chaulet et al., 2005) can be expressed as:

$$\hat{\boldsymbol{\tau}} = \frac{1}{2} \sum_{i=1}^{3} \left[ \lambda_i (\dot{\boldsymbol{\varepsilon}} : \boldsymbol{m}_i \boldsymbol{m}_i)(\boldsymbol{m}_i \boldsymbol{m}_i)^D + \lambda_{i+3}(\boldsymbol{m}_i \boldsymbol{m}_i \cdot \dot{\boldsymbol{\varepsilon}} + \dot{\boldsymbol{\varepsilon}} \cdot \boldsymbol{m}_i \boldsymbol{m}_i)^D \right], \tag{34}$$

where $\hat{\boldsymbol{\tau}}$ is the normalised stress and $\mathbf{X}^D = \mathbf{X} - tr(\mathbf{X})\mathbf{I}/3$ represents the deviatoric part of a tensor. The vectors $\boldsymbol{m}_1, \boldsymbol{m}_2, \boldsymbol{m}_3$,
are the symmetry directions of the fabric, which are assumed to be along the eigenvectors of the second-order orientation tensor
$\mathbf{A}$. The values of the terms $\lambda_i$ and $\lambda_{i+3}$ are drawn tabulated values, set to reproduce to behaviour of a micro-mechanical model
including grain interactions. In this way the GOLF aims to incorporate grain physics between the Sachs and Taylor bounds
and theoretically be more accurate. The micro-mechanical model the GOLF is constrained against (Castelnau et al., 1996a)
depends on the grain parameters $E_{cc}$ and $E_{ca}$[1]. While, as the name suggests, the GOLF was initially a linear rheology, it has
been coupled to the power law response in Eq. (24) in Pettit et al. (2007); Ma et al. (2010), although this does not affect the
normalised stress.

### 4.1.2 Rathmann's orthotropic rheology

Similarly, Rathmann and Lilien (2022) derive an alternative orthotropic rheology:

$$\dot{\boldsymbol{\varepsilon}} = \eta^{-1} \sum_{i=1}^{3} \left[ \lambda_i(\boldsymbol{\tau} : \mathbf{M}_i)\mathbf{M}_i + \lambda_{i+3}(\boldsymbol{\tau} : \mathbf{M}_{i+3})\mathbf{M}_{i+3} \right] \tag{35}$$

where $\mathbf{M}_i = (\boldsymbol{m}_j \boldsymbol{m}_j - \boldsymbol{m}_k \boldsymbol{m}_k)/2$ and $\mathbf{M}_{i+3} = (\boldsymbol{m}_j \boldsymbol{m}_k + \boldsymbol{m}_k \boldsymbol{m}_j)/2$. When $i = \{1,2,3\}$ then $j = \{2,3,1\}$ and $k = \{3,1,2\}$.
For this rheology, the viscosities $\lambda_i$ and $\lambda_{i+3}$ are constrained against a linear combination of the linear Sachs and Taylor rhe-
ologies. They use a ratio of 98.75:1.25 Sachs to Taylor weighting, so the rheology closely matches the linear Sachs rheology,
but gives a larger maximum anisotropic effect for very strong fabrics.

Rathmann and Lilien (2022) draw a distinction between the grain power law response and the macroscopic power law
response. The terms $\lambda_i$ and $\lambda_{i+3}$ depend on the grain parameters $E_{cc}$ and $E_{ca}$, the Sachs-Taylor weight, but also on the
macroscopic power exponent $n$. They also provide an unapproximated form of $\eta^{-1}$ in the context of the orthotropic rheology:

$$\eta^{-1} = A \left( \sum_{i=1}^{3} \left[ \lambda_i(\boldsymbol{\tau} : \mathbf{M}_i)^2 + \lambda_{i+3}(\boldsymbol{\tau} : \mathbf{M}_{i+3})^2 \right] \right)^{(n-1)/2} \tag{36}$$

---

[1]Gillet-Chaulet et al. (2005) refers to different grain parameters, $\gamma$ and $\beta$ (not the same as the migration recrystallization parameter $\beta$). However, these can
be expressed as: $\beta = 1/E_{ca}, \gamma = (3/E_{cc} + 1)/4$




They also provide expressions for $\lambda_i$ and $\lambda_{i+3}$ consistent with Eq. (24) and Eq. (25). We test both the unapproximated form and the rheology with approximation given in Eq. (24). For both forms for $\eta^{-1}$, the normalised stress is given by:

$$\hat{\boldsymbol{\tau}} = A(\boldsymbol{\tau} : \boldsymbol{\tau})^{(n-1)/2}\boldsymbol{\tau} \tag{37}$$

## 4.2 Other rheologies

We now review other rheologies which have been used to attempt to approximate the anisotropic effect.

### 4.2.1 CAFFE rheology

Another macroscopic rheology is the CAFFE rheology (Placidi et al., 2010). There has been some discussion in the literature over whether this rheology is truly anisotropic (Faria, 2008), as it has a co-linear relationship between stress and deformation:

$$\dot{\boldsymbol{\varepsilon}} = AE(\mathcal{D})(\boldsymbol{\tau} : \boldsymbol{\tau})^{(n-1)/2}\boldsymbol{\tau} \tag{38}$$

where

$$\mathcal{D} = 5\frac{\mathbf{A} : (\dot{\boldsymbol{\varepsilon}} \cdot \dot{\boldsymbol{\varepsilon}}) - (\mathbf{A}^{(4)} : \dot{\boldsymbol{\varepsilon}}) : \dot{\boldsymbol{\varepsilon}}}{\dot{\boldsymbol{\varepsilon}} : \dot{\boldsymbol{\varepsilon}}}, \tag{39}$$

and $E$ is a monotonic function of $\mathcal{D}$:

$$E(\mathcal{D}) = \begin{cases} (1 - E_{\min})\mathcal{D}^k + E_{\min}, & k = \frac{8}{21}\left(\frac{E_{\max}-1}{1-E_{\min}}\right), & \mathcal{D} \in [0,1] \\ \frac{4\mathcal{D}^2(E_{\max}-1)+25-4E_{\max}}{21} & \mathcal{D} \in [1, \frac{5}{2}]. \end{cases} \tag{40}$$

For an isotropic fabric, $\mathcal{D} = 1$ and $E(\mathcal{D}) = 1$, whereas for a perfect single-maximum fabric $\mathcal{D} = 5/2$ and $E(\mathcal{D}) = E_{\max}$. Therefore, expressing the rheology in terms of the normalised stress $\hat{\boldsymbol{\tau}}$, which is defined as $\hat{\boldsymbol{\tau}} = \dot{\boldsymbol{\varepsilon}}$ for an isotropic fabric only, gives:

$$\dot{\boldsymbol{\varepsilon}} = E(\mathcal{D})\hat{\boldsymbol{\tau}} \tag{41}$$

i.e. $E$ cannot be incorporated into the isotropic response. Faria (2008) shows because of this Eq. (38) is technically anisotropic, even though the relationship is co-linear.

### 4.2.2 Isotropic rheologies: Glen's and $E^*$

There have also been attempts to represent anisotropy through considering the velocity and deformation only, such as the Estar flow relation (Graham et al., 2018). This rheology can be expressed as

$$\dot{\boldsymbol{\varepsilon}} = \eta^{-1}(\dot{\boldsymbol{\varepsilon}}, \boldsymbol{u})\hat{\boldsymbol{\tau}}. \tag{42}$$

For any rheology like this, $\eta$ is not a function of the fabric. Therefore $\dot{\boldsymbol{\varepsilon}} = \hat{\boldsymbol{\tau}}$ for any fabric, and the Estar rheology will give the same fabric predictions as Glen's rheology.





### 4.3 A common equation for fabric evolution

As the above macroscopic models are not tied to either the Sachs or Taylor bound, and often aim to give a response between them, they can postulate different equations for fabric evolution. Azuma (1994) first postulated a linear blend between the Taylor and Sachs approximations:

$$\boldsymbol{v} = \boldsymbol{\omega} \cdot \boldsymbol{c} - (1-\alpha)(\dot{\boldsymbol{\varepsilon}} \cdot \boldsymbol{c} - (\dot{\boldsymbol{\varepsilon}} : \boldsymbol{cc})\boldsymbol{c}) - \alpha E_{ca}(\boldsymbol{\tau} \cdot \boldsymbol{c} - (\boldsymbol{\tau} : \boldsymbol{cc})\boldsymbol{c}), \tag{43}$$

where $\alpha \in [0,1]$, with $\alpha = 0$ corresponding to the Taylor bound and $\alpha = 1$ corresponding to the Sachs bound. Although Eq. (43) seems to suggest that as $E_{ca}$ increase the contribution to lattice rotation from the stress increases. However, if the Sachs approximation and rheology is used, Eq. (30) shows that rewriting Eq. (43) in terms of the normalised stress gives:

$$\boldsymbol{v} = \boldsymbol{\omega} \cdot \boldsymbol{c} - (1-\alpha)(\dot{\boldsymbol{\varepsilon}} \cdot \boldsymbol{c} - (\dot{\boldsymbol{\varepsilon}} : \boldsymbol{cc})\boldsymbol{c}) - \alpha\iota(\hat{\boldsymbol{\tau}} \cdot \boldsymbol{c} - (\hat{\boldsymbol{\tau}} : \boldsymbol{cc})\boldsymbol{c}), \tag{44}$$

with $\iota = E_{ca}/(0.4E_{ca} + 0.2E_{cc} + 0.4)$ (Eq. (31)). Gillet-Chaulet et al. (2006); Ma et al. (2010) used Eq. (43) with the GOLF rheology (Eq. (34)). However it should be noted that Eq. (30), i.e. Eq. (43) with $\alpha = 1$, is derived from the Sachs approximation, which results in the leading $E_{ca}$ term, and which is cancelled out when considering the normalised stress $\hat{\boldsymbol{\tau}}$. It is more accurate to consider the expressions for the normalised stress $\hat{\boldsymbol{\tau}}$ if one is postulating this relationship with a rheology other than Eq. (23), and then use these expression in Eq. (44) rather than Eq. (43). Despite this, multiple studies have used values of $\iota = 10$ in Eq (44) (e.g. Ma et al., 2010; Lilien et al., 2021). Work by Richards et al. (2021) has also used a free parameter infront of the contribution from $\dot{\boldsymbol{\varepsilon}}$. Consequently, the most general form for $\boldsymbol{v}$ which captures all previous work is:

$$\boldsymbol{v} = \boldsymbol{\omega} \cdot \boldsymbol{c} - \alpha_D(\dot{\boldsymbol{\varepsilon}} \cdot \boldsymbol{c} - (\dot{\boldsymbol{\varepsilon}} : \boldsymbol{cc})\boldsymbol{c}) - \alpha_S\iota(\hat{\boldsymbol{\tau}} \cdot \boldsymbol{c} - (\hat{\boldsymbol{\tau}} : \boldsymbol{cc})\boldsymbol{c}), \tag{45}$$

where $\iota$ is given by Eq. (31) and $\alpha_D, \alpha_S \geq 0$. This is the equation we use when modelling fabrics and testing the predictions of different rheologies. The lattice rotation $\boldsymbol{v}$ and hence fabric evolution implicitlly depends on the rheology through the appearance of $\hat{\boldsymbol{\tau}}$ in Eq. (45).

## 5 Testing the model: Application to ice sheets

In Eq. (21) and (45) we have an equation for fabric evolution which depends on a few parameters and on the anisotropic rheology used. Our goal in this work is twofold. Firstly to provide a model and set of parameters that can predict the fabrics observed in a range of locations the natural world, including ice streams. Secondly, to provide insight into which anisotropic rheology is most accurate. To tackle both these problems we apply the fabric evolution model to two locations in the Greenland ice sheet: the GRIP site, at an ice divide, and the EGRIP site, at an ice stream (Figure 3). These sites are chosen as they have extensive ice core data (Thorsteinsson et al., 1997; Stoll et al., 2021), giving all three eigenvalues of the 2nd-order orientation tensor, $\mathbf{A}$, as a function of depth. They also have similar strain rates and temperatures, meaning the same parameter set will be valid for both locations.





At each location, we find the best fit parameters (by minimising the difference between modelled and measured eigenvalues)
to observations for each rheology. Consequently, we can compare the predictions of different rheologies or models against each
other. Our hypothesis is that a model and rheology combination which actually captures the underlying physics should be able
to accurately predict the observed fabrics at both locations with the same parameter set, despite the very different deformation
conditions. A model that does not sufficiently capture the physics may be able to give good predictions at one of the locations
just by fitting a large number of parameters to the observations, but is less likely to be able to give accurate predictions at both
locations.

The fabric is simulated in a Lagrangian sense, by tracking the evolution of the fabric attached to an 'ice parcel' as it moves
through the ice sheet. As the ice parcel moves through the ice sheet, we consider how the velocity gradient, and hence $\Omega_{ij}$ and
$\dot{\varepsilon}$ evolve. These represent the inputs to the fabric evolution model, with the output being the fabric and second-order orientation
tensor to be compared to ice core observations.

We solve the fabric using a Monte-Carlo method, directly solving equations for $dc/dt$ (Eq. (45)) and $dw/dt$ (Eq. (19)) for a
large number (2000) samples, and then calculating the orientation tensors through Eq. (26).

## 5.1   Ice divide

Fabric models have commonly been compared to observations at ice divides (Castelnau et al., 1996b; Montagnat et al., 2012).
To predict fabrics at a specific ice divide and hence compare to observational measurements from drilled ice cores, we need to
approximate the deformation $\dot{\varepsilon}$ as the input to the fabric evolution model. To do this a number of assumptions must be made.
Firstly, it is assumed the core is drilled at a perfect ice divide, such that it only experiences unconfined compression:

$$\nabla\boldsymbol{u} = \begin{bmatrix} -\dot{\varepsilon}_{zz}/2 & 0 & 0 \\ 0 & -\dot{\varepsilon}_{zz}/2 & 0 \\ 0 & 0 & \dot{\varepsilon}_{zz} \end{bmatrix} \tag{46}$$

Secondly, The vertical shear rate $\dot{\varepsilon}_{zz}$ is assumed to follow a Dansgard-Johnson profile (Dansgaard and Johnsen, 1969), being
constant for the upper two-thirds of the ice divide, and then decreasing linearly to zero at the ice-bedrock interface. Once this
is assumed, the value of $\dot{\varepsilon}_{zz}$ in the upper two-thirds, $\dot{\varepsilon}'_{zz}$ can be calculated:

$$\int_0^H \dot{\varepsilon}_{zz}dz = \dot{\varepsilon}'_{zz}\frac{2H}{3} + \frac{1}{2}\dot{\varepsilon}'_{zz}\frac{H}{3} = \dot{a} \tag{47}$$

where $H$ is the ice thickness and $\dot{a}$ is the accumulation. We make a further assumption that the values for $H$ and $\dot{a}$ are unchanged
over time. For GRIP, $H = 3029$ m and $\dot{a} = 0.23$ a$^{-1}$ (Thorsteinsson et al., 1997). As we use the same parameters throughout
the entire depth of the ice divide, we are implicitly assuming that the temperature and deformation rate stay approximately
constant. The temperature at GRIP is between $-31$ and $-20°$ C above a depth of 2500 m, but increases rapidly over the
deepest 500 m to $-8°$ C at the bedrock. Therefore, we exclude this region from our analysis.



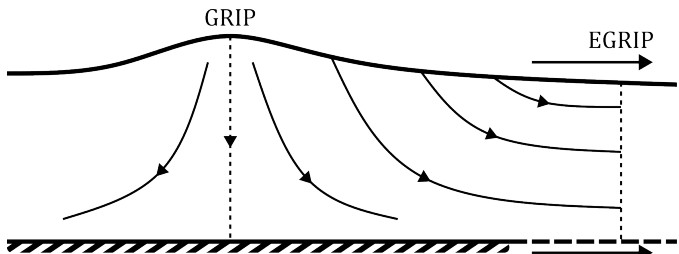

**Figure 3.** Schematic showing the location of GRIP (at an ice divide) and EGRIP (in an active ice stream). The cores drilled from GRIP all lie on the same streamline, whereas the cores drilled from EGRIP correspond to different streamlines all starting from different points upstream.

## 5.2 Ice stream

To predict fabrics at the EGRIP drill site, where ice fabric measurements are available (Stoll et al., 2021), we must estimate the three-dimensional path and deformation of the each 'ice parcel' which ends up along the vertical depth of the drilled ice core 400 (Figure 3). The ice parcel undergoes a changing history of deformations as it moves through the ice sheet.

The same methodology as Richards et al. (2023) is used to estimate the deformation. To leading order, we can assume that the ice flow around EGRIP obeys the shallow stream approximation (MacAyeal, 1989), such that the ice flows with an approximate plug flow, with negligible vertical shear. This means that the horizontal velocities (and their corresponding horizontal gradients) can be applied into the ice sheet to leading order. These horizontal velocities are found using satellite velocity data (Joughin 405 et al., 2016, 2018).

Because the horizontal velocities are applied into the depth of the ice sheet, all the three-dimensional particle paths sketched in Figure 3 follow the same surface path. Different initial locations along this surface path surface create three-dimensional particle paths that intersect the EGRIP drill site at different depths. At the starting point of each path on the surface, the vertical velocity can be found through the kinematic condition:

$$w_s = -\dot{a} + u_s \frac{\partial h}{\partial s}, \tag{48}$$

with $\dot{a}$ the accumulation rate, $u_s$ the surface velocity in the streamline direction and $\partial h/\partial s$ the change in surface height in the streamline direction, calculated from surface height data (Morlighem et al., 2017). This can then be integrated forward to give a three-dimensional path from the surface to some depth at EGRIP. For each path we iteratively choose the accumulation rate such that the the final age and depth of an ice parcel at EGRIP match the measured age-depth relationship (Gerber et al., 2021).

The measured temperature along the EGRIP core is approximately $-30°$ C to a depth of 1400 m, (Prior-Jones et al., 2021), which is as deep as the EGRIP data we have extends to and is similar to what is measured at GRIP. We assume this temperature-depth relationship is also valid upstream, along the particle paths. This means we can say the temperature is approximately constant throughout.



## 6    Results

We now move on to the results from applying described rheologies to these two locations in the Greenland ice sheet. We first present the results of the parameter fitting for each model/rheology, then show how these best parameters for each model agree with the ice core observations.

### 6.1    Parameter estimation

To re-summarise, there exists one equation for lattice rotation that contains all previously described models:

$$\boldsymbol{v} = \omega \cdot \boldsymbol{c} - \alpha_D(\dot{\boldsymbol{\varepsilon}} \cdot \boldsymbol{c} - \dot{\boldsymbol{\varepsilon}} : \boldsymbol{ccc}) - \alpha_S \iota(\hat{\boldsymbol{\tau}} \cdot \boldsymbol{c} - \hat{\boldsymbol{\tau}} : \boldsymbol{ccc}). \tag{49}$$

The deviatoric stress $\hat{\boldsymbol{\tau}}$ depends on the rheology used. Consequently, when the lattice rotation depends on the stress the fabric evolution becomes coupled to the rheology used. This means we can compare fabric predictions to ice cores to test the accuracy of different rheologies.

We review 7 different rheologies models: The Taylor and Sachs models derived from grain dynamics, as well as 5 different
macroscopic rheologies, including two variants of Rathmann's orthotropic rheology, depending on whether the unapproximated (Eq. (36)) or approximated (Eq. (24) form for $\eta^{-1}$ is used.

The rheology of a grain, and consquently the large-scale, is described by two parameters $E_{cc}$ and $E_{ca}$ controlling the increased deformation experienced by a grain in compression or shear (Eq. (1)). The parameters $\alpha_D$ and $\alpha_S$ control the contribution from deformation and stress to lattice rotation respectively in Eq (49). There are two parameters controlling
recrystallization processes: $\lambda$ and $\beta$ controlling rotational and migration recrystallization respectively. This describes fully most rheologies. Rathmann's orthotropic rheology with unapproximated form for $\eta^{-1}$ also depends on the power-law exponent $n$, and the CAFFE rheology (Eq. (38)) has unique maximum and minimum softening parameters $E_{max}$ and $E_min$.

We non-dimensionalise the two recrystallization two parameters by the macroscopic effective strain rate $\dot{\gamma} = \sqrt{\dot{\boldsymbol{\varepsilon}} : \dot{\boldsymbol{\varepsilon}}/2}$, such that $\lambda = \tilde{\lambda}\dot{\gamma}$, $\beta = \tilde{\beta}\dot{\gamma}$, as in Richards et al. (2022).

While there are a large number of parameters, there are a number of physical constraints available to reduce the parameter space. The Taylor model requires $\alpha_D = 1, \alpha_S = 0$ in Eq. (49) (and hence does not involve a rheology), while the Sachs model requires $\alpha_D = 0$, $\alpha_S = 1$. The other 4 rheologies place no restriction on the values of $\alpha_D$ and $\alpha_S$. Furthermore, there is no evidence to suggest that compression is easier or harder in the direction of the $c$-axis or perpendicular to it (Gillet-Chaulet et al., 2005), which means $E_{cc} = 1$. For $E_{ca}$, Rathmann and Lilien (2021) finds a value of $10^4$ best reproduces the expected
polycrystal enhancement of 10. However we use $E_{ca} = 10^2$ as this is the maximum value the GOLF rheology is constrained against, and we aim to make comparisons between rheologies. This has only a minimal effect on the results here.

The GRIP ice divide (above a depth of 2500 m) and the EGRIP ice stream are at $T < -20°$ C. At these temperatures migration recrystallization does not have a significant effect on the fabric (Craw et al., 2018; Qi et al., 2019). Therefore we neglect it ($\beta = 0$).

As it is the normalised stress which appears in Eq. (45), most models do not depend on the value of $n$ used. The exception is the unapproximated form of Rathmann's rheology, for which we use $n = 3$. Furthermore, when rheologies have internal



**Table 1.** Comparison best fit parameters of different model and rheologies. Parameters in italics were chosen by fitting, blank entries mean the paramater is not included in the model.

| Model/rheology | $\alpha_D$ | $\alpha_S$ | $\tilde{\lambda}$ | $\tilde{\beta}$ | $E_{cc}$ | $E_{ca}$ | $n$ |
|---|---|---|---|---|---|---|---|
| Taylor | 1 | 0 | *0.03* | 0 | | | |
| Sachs (Eq. (23)) | 0 | 1 | *0.32* | 0 | 1 | $10^2$ | |
| GOLF (Eq. (34)) | *0* | *1.15* | *0.45* | 0 | 1 | $10^2$ | |
| Rathmann (Eq. (35)) w/ Full $\eta$ | *0* | *1.47* | *0.49* | 0 | 1 | $10^2$ | 3 |
| Rathmann (Eq. (35)) w/ $\eta = (\boldsymbol{\tau} : \boldsymbol{\tau})^{(n-1)/2}$ | *0* | *1.1* | *0.34* | 0 | 1 | $10^2$ | |
| CAFFE (Eq. (38)) w/ $E_{\max} = 10$, $E_{\min} = 0.1$ | *0* | *1.1* | *0.36* | 0 | | | |
| Estar/Glen (Eq. (42) | *2.6* | 0 | *0.32* | 0 | | | |

parameters we use those suggested by their authors. Rathmann's rheologies depend on a Sachs-Taylor weight, which we use 98.75:1.25 following (Rathmann and Lilien, 2021). The CAFFE model depends on parameters $E_{\max}$ and $E_{\min}$, for which we use the values of 10 and 0.1 respectively, as is suggested in (Placidi et al., 2010). While the CAFFE model does not depend on

$E_{cc}$ and $E_{ca}$, we use the same value of $\iota$ in Eq. (45) for ease of comparison.

For the Sachs and Taylor models, this leaves only $\tilde{\lambda}$ to be constrained. For the GOLF, Rathmann and CAFFE rheologies 3 parameters must be determined: $\alpha_D$, $\alpha_S$ and $\tilde{\lambda}$. Glen's rheology has only 2 free parameters: $\alpha_D$ and $\tilde{\lambda}$ because $\dot{\varepsilon} = \hat{\tau}_{ij}$.

For all models/rheologies the parameter space is explored and a set of best parameters are chosen (Table 1), aiming to give the best fit for both locations. There is not one unique set of parameters which gives the unambiguous best fit for each model,

but the parameters do show the best possible fit for each model/rheology.

The fabric predictions from the parameters shown in Table 1, for each model/rheology, are compared to observations from the GRIP ice divide and the EGRIP ice stream.

## 6.2   GRIP ice divide

At the ice divide we assume the fabric undergoes unconfined compression as it moves vertically down. The measured largest

eigenvalue from different depths at GRIP is shown in Figure 4 as crosses (Thorsteinsson et al., 1997). The same data points are shown on both subplots. As the fabric at an ice divide has rotational symmetry around the vertical axis, the two smaller eigenvalues are equal. Consequently, we only plot the largest eigenvalue.

Figure 4 also shows the results from the suite of models/rheologies tested, plotted as solid lines. The figure has been split into two subplots to aid readability. The tensor rheologies are on the right-hand subplot. Because of the way we chose parameters,

first finding a set that agreed well with this case and then improving the fit for the ice stream, all the models in the right subplot agree well with observations here. There is very little difference between the best predictions of the Sachs, GOLF and Rathmann w/ Petit models. The unapproximated Rathmann rheology has marginally worse agreement with ice cores above a depth of 1500 m.



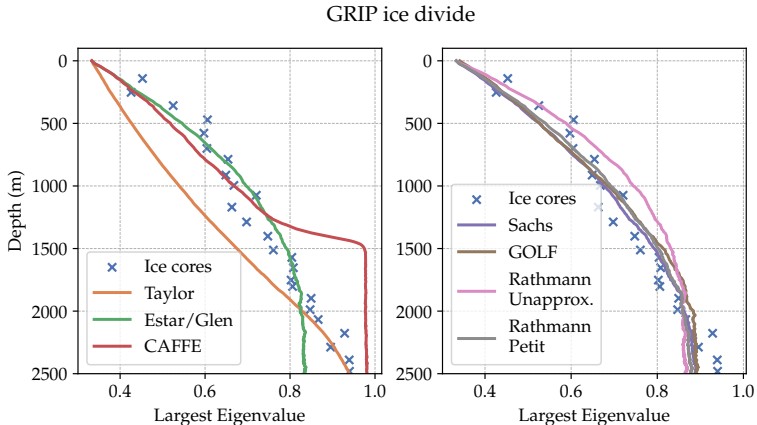

**Figure 4.** Evolution of largest (vertical) eigenvalue of the second-order orientation tensor **A** at the GRIP drill site (an ice divide) with depth. Only the largest eigenvalue is shown as the fabric has rotational symmetry about the vertical, so the other two eigenvalues are equal. Measurements from ice cores are plotted as crosses, showing an increase with depth. Model results are plotted as lines, and have been split across two subplots to aid readability. The right hand plot shows the tensorial rheologies.

In the left subplot of Figure 4, the Glen/Estar rheology with $\alpha_D = 2.6$, $\alpha_S = 0$ and $\tilde{\lambda} = 0.32$ also gives good agreement with the observed eigenvalues. For this rheology, this rheology $\hat{\tau} = \dot{\varepsilon}$. Therefore this parameter set could equivalently be expressed as $\alpha_D = 0$, $\alpha_S = 1.06$, $\tilde{\lambda} = 0.32$, for the same value of $\iota = 2.46$ in Eq. (45), as in the other models. This is a similar parameter set to that found for the Sachs and Rathmann Petit models, highlighting that for this simpler case of the ice divide, it is not necessary to have an anisotropic rheology contributing to the fabric evolution to get reasonably accurate predictions.

The CAFFE model exhibits a sharp increase of the largest eigenvalue to $\approx 1$ at a depth of around 1500 m. This is not seen in observations for GRIP but has been observed to an extent at other ice divides (Montagnat et al., 2012). We find that this sharp increase is sensitive to the value of $E_{\min}$, though detailed exploration of this is beyond the scope of this work.

## 6.3 EGRIP ice stream

The flow at an ice stream is much more complex than an ice divide. Consequently, at each depth at the EGRIP ice core the ice there has experienced changing deformation history. In Figure 5 we show the observed eigenvalues at EGRIP with depth as blue dots. Each eigenvalue is shown in a separate column. The data is also reproduced across each row to aid readability. Close to the surface at a depth of around 50 m, the observed eigenvalues at EGRIP have one larger eigenvalue (the vertical) and two smaller, indicating a single-maximum fabric produced as snow compresses into ice. We do not model this effect. Below this, the fabric is produced by ice stream deformation, and is characterised by one eigenvalue $\approx 0$, and two larger eigenvalues, corresponding to a girdle fabric. This fabric is fairly invariant with depth below 550 m, with some noise.

We also show the results from the suite of models/rheologies tested in Figure 5. The first row shows results from the Taylor, Glen and CAFFE models, and the second row shows the results from the tensorial rheologies. We do not include the effect of





firn in our analysis, which significantly affects the observed fabric above a depth of around 550m. Hence, we only compare the models to observations below a depth of 550 m.

The Taylor model does not give good agreement with the EGRIP observations, showing one much larger eigenvalue of $\approx 0.8$
over the depth we are interested in (below 550 m). The best parameters with Glen's rheology and the GOLF rheology also do not agree, not predicting that the smallest eigenvalue is much larger than what is observed. From this figure, the CAFFE rheology, Sachs rheology, and both versions of Rathmann's rheology give reasonably good agreement with observations, predicting a smaller eigenvalue close to zero and two larger eigenvalues.

To provide a more detailed comparison, Figure 6 shows a comparison of modelled to observed eigenvalues in a depth-
averaged sense. Observed eigenvalues from EGRIP are averaged over a depth of 550 to 1425 m, and plotted as dashed lines with the standard deviation shown by the shaded region. In this figure, the colour coding simply shows which eigenvalue (largest, middle or smallest) is being referred to. Then for each model/rheology ($x$-axis), we average the modelled eigenvalues over the same depth and plot them as as crosses, with the same colour coding as observations. Distance of the crosses from the respective dashed lines then allows us to assess agreement of models with observations in this deeper area of the ice core.

Figure 6 shows that the CAFFE, Sachs, and both Rathmann rheologies all have their predictions for the largest two eigen-values within 1 standard deviation of the mean observed value at EGRIP. They also all have the smallest eigenvalue close to zero, with the Sachs model being closest to observations. This figure also shows that while Glen's and the GOLF rheologies with their respective best parameters are less accurate, they are both significantly more accurate than the Taylor model.

Previously, we have limited our analysis to solely the eigenvalues. We now extend this to examine the pole figures - showing
the complete orientation distribution function (ODF) - directly in Figure 7. The ODF represents the probability of finding a grain at each orientation. Brighter regions indicate a high probability of a grain being orientated towards that orientation. The space of possible orientations exist on the surface of a sphere. There is also no possibility of distinguishing a grains top from bottom, so the ODF is symmetrical between its north and south hemispheres. Consequently, we plot the ODF over a single hemisphere, looking vertically downwards on the fabric. Figure 7 shows two pole figures observed from EGRIP on the left.
While we have eigenvalue data over a large depth range, we only have pole figures for these two depths. There is variance between these two depths. The corresponding eigenvalues to these EGRIP pole figures are shown in Figure 5 as black crosses, showing that these two pole figures represent end members of the variance seen at EGRIP. Despite this, they both show a girdle fabric pattern, indicating most grains have $c$-axes pointed along the bright band.

We also show pole figures for each model in Figure 7. The Taylor model had one large eigenvalue and the other two $\approx 0$.
This corresponds to what is seen in the pole figure: a cluster in only 1 direction, which is repeated due to the symmetry of the ODF. Glen's rheology and GOLF rheologies both predicted the smallest eigenvalue was larger compared to observations, and this corresponds to a generally too diffuse pole figure, which was also seen in (Richards et al., 2023), which was the previous state-of-the-art for modelling ice stream fabrics.

The remaining models all provide improvements on this previous state-of-the-art. The pole figures for the Sachs model and
the two versions of Rathmann's rheologies lie within the variance of what is seen in the EGRIP pole figures, predicting girdle fabrics with a slight bias towards grains being orientated towards the edges of the pole figure. The CAFFE model also predicts



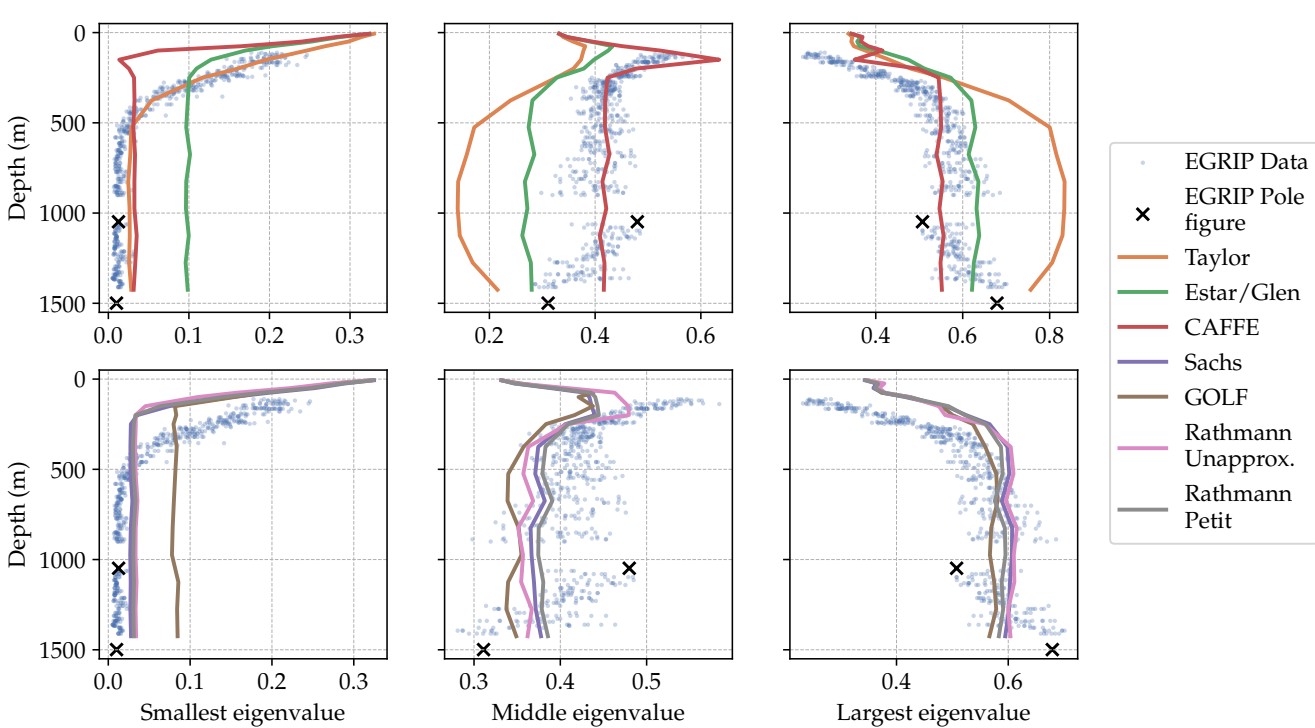

**Figure 5.** This figure shows the measured eigenvalues at EGRIP, plotted as dots, alongside the results for each model/rheology (lines). Each column represents a single eigenvalue (smallest, middle, largest) and the figure has also been split over two rows to aid readability, with the tensorial rheologies in the second panel. Highlighted with black crosses are the eigenvalues of the pole figures shown in Figure 7.

a girdle fabric, though the detail features of the pattern in Figure 7 show it is slightly different from what is seen at EGRIP, with peaks closer to the centre of the figure. These detail features can only be examined through looking directly at the pole figures.

These results are summarised in Figure 8. Here we show the summed difference between the modelled and observed eigen-
values for each model/rheology. This error is then normalised against the results for Glen's rheology (so the results for Glen are by definition 100%). This again highlights that the Sachs and Rathmann w/ Petit models have the closest agreement for both the ice divide and ice stream. Figure 8 also shows how the anisotropic rheologies provide the significant improvements in fabric predictions for the ice stream case (EGRIP) only.

## 6.4 Sensitivity to a changes in the ice stream over time

In our previous analysis, as explained in Section 5.2, we use the present-day satellite velocity data to trace ice parcels upstream. However tracing a particle path upstream also means tracing a particle path back in time. Consequently, this method implicitly



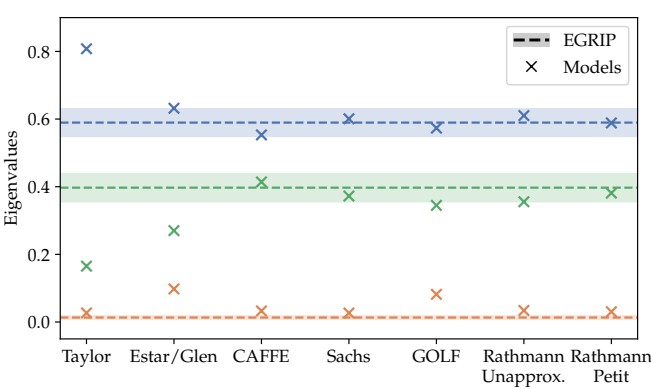

**Figure 6.** This figure shows, for each model/rheology, the mean eigenvalues over a depth of 550 to 1425 m. Colours indicate the different eigenvalues. The horizontal line shows the mean of the observed eigenvalues over this depth, the shaded region corresponding to $\pm 1$ standard deviation. Along the $x$-axis, the mean eigenvalues for each model are plotted as crosses.

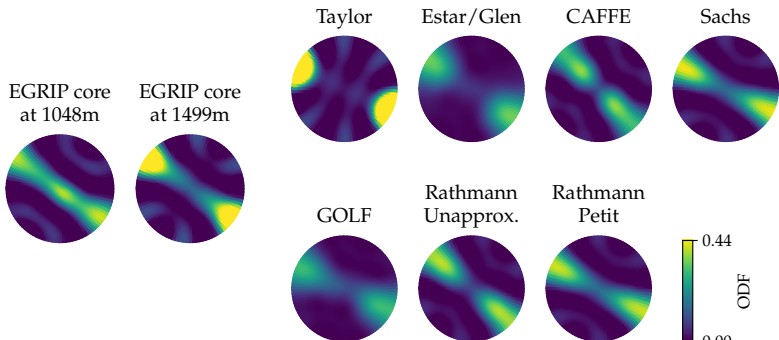

**Figure 7.** Comparison of pole figures, for all models and measured from an ice core at two depths, for which the corresponding eigenvalues are highlighted in Figure 5. The pole figure shows a hemisphere of the orientation distribution function, which is sufficient to show all the information. Pole figures are shown looking vertically downwards, with North at the top. For example, the observed fabric at EGRIP at 1048 m shows that almost all grains lie close to a plane normal to the North-East direction.

assumes the ice velocity field has not changed over the past 16,000 years (the time it takes for ice to travel from the furthest point upstream to EGRIP).

Recent work has raised the possibility that the North-East Greenland Ice Stream (NEGIS) may be relatively young, existing only for 2000 years (Jansen et al., 2024), before which no ice stream existed here. To test the robustness of our results to a 'young NEGIS', we assume that before a certain time, there was no ice stream, i.e. the ice was frozen to the bed. Consequently, the deformation would be dominated by vertical shear and a single-maximum would be produced, as is observed at locations like NEEM and NGRIP (Montagnat et al., 2014). Ice cores from these locations reveal a vertical single maximum with a largest





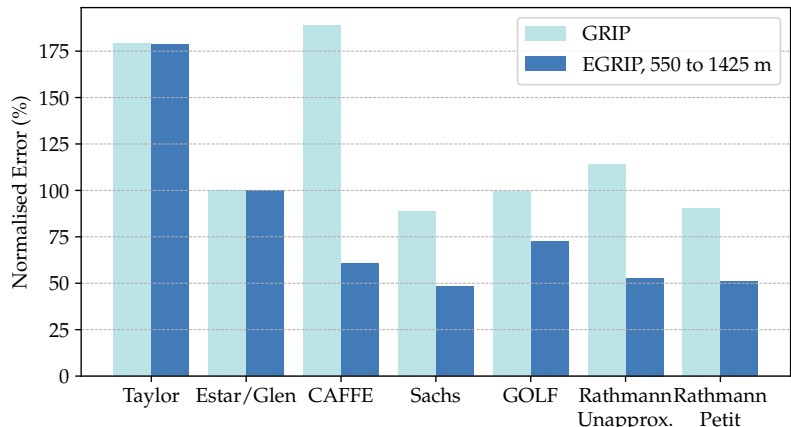

**Figure 8.** Normalised (relative to best parameters with Glen's rheology) error for both GRIP and EGRIP. This is calculated by considering the sum of the difference between observed and measured eigenvalues at each depth.

eigenvalue of around 0.75 over the depth we examine. Therefore we assume this as the pre-existing fabric over the whole depth.

The ice stream deformation is then switched on at a certain time ago, and the fabric evolution is calculated from then onwards, with the vertical single-maximum as the initial fabric.

We explore this in Figure 9. This again shows how the average eigenvalues evolve as in Figure 6, but this time depending on the age of the NEGIS. The results are shown only for the Sachs model with the parameters shown in Table 1, though the trend does not change between models. Figure 9 shows that if the ice stream was extremely young (e.g. 100 years) the fabric would

still be a vertical single maximum. If the NEGIS is was only 2000 years old, our results change slightly: a girdle fabric is still predicted, but the eigenvalues are different, consequently the best parameters would be slightly different. However, even for a young NEGIS of only 3000 years or older, the results are unchanged. Consequently, we can say our results are robust to an ice stream age of 3000 years or older. If the ice stream is found to be only 2000 years old, our best fit parameters change, but the general conclusions and ranking of the different models does not. Reproductions of Table 1, and Figures 4, 6 and 7 for a 2000

year old ice stream can be found in the supplement.

## 7 Discussion

In the introduction, we set out two main problems facing accurate modelling of viscous anisotropy in ice sheet models. Firstly, a model for fabric evolution which can give accurate predictions of fabrics observed in the natural world, specifically in ice streams, is missing. Secondly, there has been no way to test whether an anisotropic rheology is accurately representing the

anisotropic effect, leading to a large number of competing rheologies.

In the first part of this paper (sections 2, 3 and 4) we examined a range of existing models of anisotropy in ice, starting from the dynamics of an ice grain. We showed that previous models for fabric evolution (e.g. Azuma and Goto-Azuma, 1996; Gillet-





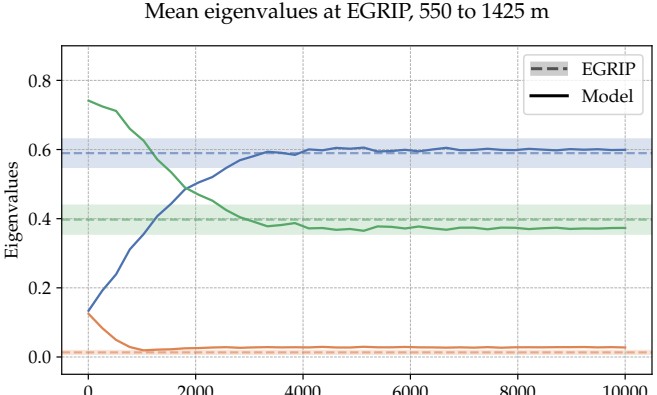

**Figure 9.** This figure shows how the results for the Sachs model change at EGRIP, for an ice stream that only started x000 years ago ($x$-axis). Plotted is the modelled mean eigenvalue as the solid line. For comparison, the mean and standard deviation of the observed eigenvalues at EGRIP is also shown as in Figure 6, as the dotted and shaded region respectively. Colours indicate the eigenvalue (i.e. smallest, middle, largest).

Chaulet et al., 2006; Richards et al., 2021; Lilien et al., 2021) can be expressed as a common equation (Eq. (45)) depending on a few parameters and also the rheology used. Therefore, the choice of anisotropic rheology becomes part of the fabric evolution
model.

In Section 5 and 6 we applied these models to two locations in the Greenland ice sheet: an ice divide and ice stream, resulting in significantly more accurate results than previous work (Richards et al., 2023). We found the best fit parameters for each rheology. These best fit parameters (Table 1) suggest that lattice rotation due to stress, consistent with the assumption that all grains are experiencing the same stress (Eq. (30)), is the most physically accurate. Of the rheologies examined, we found
that the rheologies with best agreement to observations in both the ice stream (Figure 5) and ice divide (Figure 4) where the Sachs rheology (Eq. (23)) and Rathmann's rheology (Eq. (35)). Both these rheologies describe anisotropy through a fourth-rank tensor, rather than attempting to represent it through some scalar enhancement factor. They both also assume the ice grains are experiencing approximately the same stress.

### 7.1 How is the ice grain deforming?

The results we have shown support the case for ice grains deforming closer to the Sachs hypothesis - that all grains experience the same stress - than the Taylor hypothesis - that all grains experience the same deformation. We conclude this because of these two models derived directly from grain dynamics, the Sachs predicts eigenvalues much closer to observations, for both the divide (Figure 4) and the ice stream (Figure 5), than the Taylor model. The Taylor model is also not able to predict the girdle fabric observed in the ice stream (Figure 7). Furthermore, we observed that the Sachs model had the closest agreement
to observations of all the examined models in this work. This is despite the fact that the other rheologies have more free



parameters, which would be expected to result in better predictions. Indeed, for the anisotropic rheologies where $\alpha_D$ and $\alpha_S$ (controlling lattice rotation due to deformation and stress respectively) can take any value, the best predictions come from choosing values close those required by the Sachs model.

One caveat to this is the variable amount of diffusion/Browninan motion (controlled by $\lambda$) required to give accurate predictions. If there is no diffusion acting on the fabric ($\lambda = 0$), the Sachs model vastly over predicts the fabric strength at both locations. This diffusional process is meant to mimic rotational recrystallization. Observations of grain boundaries suggest some rotational recrystallization is indeed occurring in the EGRIP (Stoll et al., 2021). However it is possible that the some amount of the modelled diffusion of the fabric is acting to ameliorate an over-prediction of lattice rotation from the Sachs hypothesis alone. Castelnau et al. (1996b); Thorsteinsson (2002) showed that including nearest neighbour interactions on grains can have a diffusional-like effect on the fabric. Consequently, we are cautious about using this model to infer rates of rotational recrystallization.

## 7.2 What is needed for accurate fabric modelling?

Previous work including only lattice rotation due to deformation has indeed succeeded at ice divides (Lilien et al., 2021; Richards et al., 2023). Yet it is worth noting that at divides the deformation is steady over time, and the observed fabric is simple - a single maximum. This makes it fairly easy for a model with a number of free parameters to fit to observations. In contrast, ice streams have a changing deformation history and the observed fabric at EGRIP is more complex. Models which take only lattice rotation due to deformation into account have not been able to accurately model this fabric (Gerber et al., 2023; Richards et al., 2023). Conversely, we observe that lattice rotation due to stress rather than deformation is key, combined with an anisotropic rheology like the Sachs or Rathmann's rheologies (Figure 7). The parameter fitting in Section 6.1 finds that for all models with free parameters controlling both lattice rotation due to deformation ($\alpha_D$) and due to stress ($\alpha_S$), the best agreement comes from lattice rotation due to stress only (Table 1).

Furthermore, the fabric evolution model is only able to accurately predict the fabric observed at EGRIP is it is linked to either the Sachs or Rathmann rheologies. What these rheolgies have in common is that they a) have a fourth-rank tensor description of the anisotropy and b) assume the grains deform with approximately the same stress. The other models are not able to predict the observed fabrics at EGRIP for any set of parameters chosen. The Sachs and Rathmann rheologies more accurately predict the form of the stress tensor, and are consequently able to accurately predict the eigenvalue evolution and correct pattern for both an ice stream and an ice divide, two very different deformation conditions. Consequently, a fabric model with either the Sachs or Rathmann w/ Petit rheology , along with the parameters shown in Table. 1 can be applied more generally and can be used in ice sheet models, at least for low temperature conditions ($T < -20°$ C) which apply to GRIP and EGRIP.

## 7.3 Which rheologies are most appropriate?

Including the effect of fabric on the large-scale flow requires two parts. Firstly, accurate modelling of fabric evolution as we have discussed above. Secondly, it requires an accurate anisotropic rheology. We recommend the Rathmann rheology. By incorporating the rheology into the fabric evolution model, we are able to compare the accuracy of different rheologies and





provide a test of them for the first time. All the anisotropic rheologies provide an improvement over Glen's rheology. For each

rheology, we have chosen the set of parameters which minimise the difference between the modelled and observed eigenvalues. Despite this, only the Sachs and Rathmann rheology with Petit approximation are able to predict the observed girdle fabric at EGRIP, and have the minimum difference between the modelled and the observed eigenvalues.

Despite providing accurate fabric predictions, the Sachs rheology under-predicts the maximum possible effect of fabric on viscosity. Experimental tests (Pimienta and Duval, 1987), have shown a vertical single maximum fabric has approximately 10

times higher viscosity in the vertical direction compared to an isotropic fabric (i.e. a fabric with the grains randomly orientated). However, the Sachs rheology (with linear grain behaviour as used here) predicts a maximum ratio in viscosity between these two fabrics of only 2.5 (Gagliardini et al., 2009). This effect does not impact fabric predictions due to our use of a normalised stress throughout (Eq. (28)).

Fortunately, this disadvantage does not affect Rathmann's rheology, with either the unapproximated form of $\eta$ or the Petit

approximation. To give the correct maximum enhancement with Rathmann's rheologies, an $E_{ca}$ value of $10^3$ should be used, along with the Sachs-Taylor weight used here. As is seen in Fig. S9 of the supplement, there only a small effect of increasing $E_{ca}$ above 100 on fabric predictions, as its effect on the eigenvalues asymptotically approaches some value as $E_{ca} \to \infty$. Therefore, we recommend using Rathmann's rheology in ice-sheet models.

It is surprising that the GOLF rheology performs poorly. The GOLF reproduces the response (through tabulated values) of

the VPSC model, a microscopic model which includes grain interactions, and should theoretically be more accurate than the Sachs or Taylor bounds, or a linear combination of them. A possible explanation for this is that the underlying VPSC model is outdated: Montagnat et al. (2012) notes that this VPSC model cannot accurately predict fabrics in simple shear, which occurs in the ice stream. This has been improved upon (Signorelli and Tommasi, 2015), so it would be interesting to see how the GOLF performs with an updated underlying model.

Finally, the CAFFE model (Placidi et al., 2010) gives some interesting predictions. It gives good predictions for the eigenvalues at EGRIP. However, for the ice divide case it predicts a sharp increase of the largest eigenvalue at 1500m which is not seen in other models or the ice core observations. This sharp increase is sensitive to the parameter $E_{min}$ (see Fig. S12 of the supplement) whereas Placidi et al. (2010) claim this parameter can be freely chosen between 0 and 0.1. Nevertheless, it provides a significant increase in accuracy for fabric predictions over Glen's rheology, and is easier to incorporate into ice-sheet

models than a tensor flow relation.

## 7.4 Limitations of this analysis

In this contribution, we have constrained a general model for fabric evolution against an ice divide and an ice stream. This has shown that lattice rotation due to stress, where the stress is found through an anisotropic tensor rheology, is necessary to accurately predict fabrics, particular those observed in the ice stream location, where previous models could not provide

accurate predictions. One limitation of this work is that we have only explored one ice divide and ice stream each. There exist other observations from different divides (see Faria et al. (2014b) for a review). Fortunately, the observed fabrics at different ice divides are generally similar, so our results applied to GRIP should also apply elsewhere.



Unfortunately, there is only one ice core drilled from an active ice stream: EGRIP. Recent advances with radar (e.g. Young et al., 2021) and seismics (e.g. Lutz et al., 2020) provide some information on fabrics at other ice streams, however they do not
provide the full eigenvalue and pattern information with depth that is available from an ice core. Now we have a fabric model we believe is accurate, it can be applied to locations where radar and seismic measurements are available to see if it agrees with these observations. Ideally, there would be another full ice core available from an active ice stream.

We have also not considered the effect of temperature. In fact, while the parameters we have found give good predictions for both EGRIP and GRIP, and can be applied to low temperature regions ($T < -20°$C), further work is required to constrain these
parameters at higher temperatures. At the lattice scale, far too small to resolve in this model, all the processes are dependent on temperature. At the larger scale, this means the parameters controlling recrystallization processes and lattice rotation will be temperature dependent. Migration recrystallization is known to be highly temperature dependent (Richards et al., 2021). However, the comparison to GRIP and EGRIP only covers a temperature range of $-40°$ to $-20°$ C. Consequently, the parameters shown do not cover the highest temperature seen in ice sheets. Although we have neglected migration recrystallization in this
study, it is expected to be significant at higher temperatures (Faria et al., 2014b; Richards et al., 2021). These high temperature regions include ice shelves and the ice-bedrock interface, both areas of critical importance to the large-scale flow.

As discussed in Section 6.4, as part of this analysis we have assumed the deformation around EGRIP is unchanged going back around 16,000 years. However, there is open discussion in the literature over whether the ice stream at EGRIP is a recent formation. Some recent work (Jansen et al., 2024) has suggested it may be as young as 2000 years. We have shown in Figure
9 that the results used here are unchanged for an ice stream age as young as 3000 years. For an age of 2000 years, the best fit parameters are changed slightly (Table S1), but the overall conclusions and ranking of best rheologies is unchanged.

## 8   Conclusions

In the introduction, we mentioned that there are currently two main problems in modelling anisotropy: (i) what is the correct physics for modelling fabric evolution and (ii) how can we know if competing anisotropic rheologies in the literature are
accurately representing the anisotropic effect. In this work we have made a step towards addressing both these questions. This is done by first showing that many previous fabric evolution models for ice can be written as one common equation, subject to different rheologies and parameters. These different models and rheologies are then tested against observations and their results compared.

We find that only with certain physical assumptions can accurate predictions of ice fabrics be obtained across a range of
natural conditions, including predicting the girdle fabric observed at EGRIP. The physics required is lattice rotation due to stress, where the stress is calculated using an anisotropic rheology. This is either the Sachs rheology (which assumes all grains experience the same stress) or Rathmann's rheology (Rathmann and Lilien, 2022) (which is close to this approximation). Alongside this, it is necessary to include a diffusive process acting on the fabric modelling rotational recrystallization.

We also provide a comparison of competing anisotropic rheologies, by testing which rheologies give the best fabric predic-
tions. As mentioned above, the Sachs rheology and Rathmann's rheology give the best predictions, suggesting they accurately

represent the anisotropic effect. However, as the Sachs rheology under predicts the maximum softening observed for strong fabrics, we suggest the rheology of Rathmann and Lilien (2022) is most physically accurate overall.

While further work is needed to find the correct parameters for fabric evolution at higher temperatures (where migration recrystallization will become important) this works provides significant insight into the physical assumptions to use when
modelling anisotropy in ice-sheet models.

*Code and data availability.* The EGRIP eigenvalue data is available on PANGAEA, https://doi.org/10.1594/PANGAEA.949248 (Weikusat et al., 2022) . The GRIP eigenvalue data was digitised from Thorsteinsson et al. (1997). Code which can reproduce the figures in this paper is available at https://doi.org/10.5281/zenodo.13866542 (Richards, 2024), which contains code to extract the relevant data from velocity maps of Greenland at https://doi.org/10.5067/QUA5Q9SVMSJG (Joughin et al., 2016) and from surface height data, https://doi.org/10.5067/
FPSU0V1MWUB6 (Morlighem, 2022). The EGRIP age-depth relationship is taken from the supplement of Gerber et al. (2021). The pole figure data from EGRIP is available at https://doi.org/10.5281/zenodo.8015759 (Stoll and Weikusat, 2023). The Monte-Carlo based solver is available at https://github.com/dhrichards/mcfab.

*Author contributions.* DR and EM conceptualised the research, DR also developed the methodology, software and figures, and drafted the paper. EM supervised and reviewed and edited the draft. SPe and SPi conceptualised the ice parcel tracing method and reviewed and edited
the draft.

*Competing interests.* Elisa Mantelli is an editor for The Cryosphere.

*Disclaimer.* The views and opinions expressed here are those of the author(s) only and do not necessarily reflect those of the European Union or the European Research Council Executive Agency. Neither the European Union nor the granting authority cance behaves differently depending on its crystal orientation, but how this affects its flow is unclear. We combine a range of previous models into a common equation
to better understand crystal alignment. We tested a range of previous models on ice streams and divides, discovering that the best fit to observations comes from a) assuming neighbouring crystals have the same stress, and b) through describing the effect of crystal orientation on the flow in a way that allows directional variation. be held responsible for them.

*Acknowledgements.* This research was supported by the Australian Research Council Special Research Initiative, Australian Centre for Excellence in Antarctic Science (Project Number SR200100008). EM was supported by the European Union (ERC-2022-STG, grant no.
101076793) and by funding from the Helmholtz Association. We also thank Nicolas Stoll, Ilka Weikusat and the EGRIP team for gathering the fabric data referenced in this manuscript. EastGRIP is directed and organised by the Centre for Ice and Climate at the Niels Bohr Institute, University of Copenhagen. It is supported by funding agencies and institutions in Denmark (A. P. Møller Foundation, University of




Copenhagen), the USA (US National Science Foundation, Office of Polar Programs), Germany (Alfred Wegener Institute, Helmholtz Centre for Polar and Marine Research), Japan (National Institute of Polar Research and Arctic Challenge for Sustainability), Norway (University of Bergen, Trond Mohn Foundation), Switzerland (Swiss National Science Foundation), France (French Polar Institute Paul-Émile Victor, Institute for Geosciences and Environmental Research), Canada (University of Manitoba) and China (Chinese Academy of Sciences, Beijing Normal University)





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
