# Peer review of "A unified framework for large-scale fabric evolution models and anisotropic rheologies"

_EGUsphere, 2024_

## Referee Comment (RC1)

**Review of "A comparative study of fabric evolution models and anisotropic rheologies" by Richards et al.**

This paper presents a unified framework in which most recent, large-scale ice-crystal fabric modeling can be considered. The challenge in constructing the framework was mainly in the non-collinearity between stress and strain rate with anisotropic rheologies, which bleeds into the fabric evolution. The unified framework clarifies this effect of anisotropy and allows for using quite a few different rheologies in a unified model of fabric evolution. This unified fabric model is used to try to reproduce data from the GRIP and EGRIP ice cores, with 7 different rheologies, and parameters tuned for each rheology to fit observations. Several of the rheologies allow the model to do a good job reproducing the fabric observations. From these results, the authors recommend one of these rheologies for use going forward.

**General comments**
This is an interesting study, and I enjoyed reading it. The writing is good though occasionally repetitive, and the figures are well made. While this framework is not a huge change from the existing descriptions of lattice rotation, the use of the normalized stress to support this general framework is a nice addition, and the framework provides clarity on how parameters relate between models. Overall, I think it is valuable to be able to compare these rheologies in a common framework for fabric evolution, and I am quite impressed by the number of rheologies compared.

However, I think the authors should temper and reframe the latter part of work; this is a convoluted way to test rheologies, and I am not convinced that it should be used for this purpose. Contrary to the claim in the introduction, it has in fact been possible to test rheologies—we have laboratory deformation tests against which to compare rheologies, and this has been done by one of the studies considered extensively within the present manuscript (Rathmann and Lilien, 2022b). Comparing to deformation tests is a much, much more direct test of the rheologies than feeding them through a fabric evolution with tuned parameters, uncertain flow history, and simple assumptions about flow (shallow shelf or Dansgaard-Johnsen). As the authors point out late in this work, the Sach's hypothesis cannot produce the extent of enhancement observed with real fabrics by about an order of magnitude—that is a lot more straightforward than sending it through a fabric model! This work tuned multiple parameters, one of which carries physical meaning (lambda), to make these rheologies reproduce observed fabrics—without knowing that the value of lambda is reasonable, how do we know that a rheology gets the right fabric for the right reason, rather than due to this tuning? I do find the comparison of fabric predicted using Taylor, Estar, CAFFE illuminating, since I would have expected them to do better, but I am not convinced that this is a meaningful way to distinguish between the other rheologies considering the tuning and uncertainty. As a result, I would like to see the conclusions tempered. In terms of rheologies, I think all I conclude is that most of them can be tuned to do a pretty good job reproducing observations of fabric—not that one of Sachs, Rathmann, or GOLF is superior *based on the present work*. I think the work is publishable in *The Cryosphere* with relatively minor changes if this rheology recommendation is tempered/removed.

**Specific comments**
First, it is Dr. Pettit, not Petit.

I think there are implicit assumptions about collinearity of stress and strain that might confuse the reader. At a minimum, the assumptions should be stated. Equation 6 does not appear to be a result of Equations 2 and 5, and I think it involves more assumptions that the authors are crediting. Clearly plugging equation 2 into 5 does not straightforwardly lead to 6—perhaps I am missing a complicated derivation demonstrating this, but I do not think that I am. Rather, I think that this is essentially an assumption about the collinearity of the strain rate and stress. If I understand, 6 is essentially an alternative to 5, and then by combining them we get something like the fabric evolution equation in 44? Considering the conclusions of the paper, this does not really make sense—the reader needs a full accounting of whether Eq. 6 is alternative to the strain-rate based model (as I read it) or an equivalent formulation as currently implied. Relatedly, I think the introduction of normalized stress is muddled, which might confuse the reader about what is colinear under that assumption. In essence, I think Eq. 28 is essentially a property of 29 that emerges for isotropy, but it took me a long time to see this. If we had a clear definition of normalized stress early, that was then consistent across rheologies, I would have an easier time conceptualizing the results.

It does not really make sense to try to fit these rheologies to data when they are mis-constrained at the surface. That is to say, it appears that the models struggle at EGRIP in part because the shallowest available data are not that close to isotropy. I suggest assuming that reorientation processes differ in the firn, justifying the use of the shallowest measurement of fabric as the initial state for the model. In this way, the misfit at the top of the ice sheet, which I do not think is due to the models per se, can be avoided, and they can be compared over more depths.

I do not see why the authors exclude migration recrystallization under the Taylor hypothesis—including it by using the strain rate to calculate $s'$ seems like a reasonable alternative to allow better comparison, and presumably would be very easy to do. Particularly since it doesn't need to be coded up, as beta is taken to be zero, this is no work to include.

On the topic of migration recrystallization, there should be discussion on how the possibility of low temperature, high stress/strain migration recrystallization might affect the results. There is some work that suggests such a possibility (see the Faria 2014 reviews), so its effect on the results here should be considered. I do not necessarily see a need to tune Beta, but discussion is warranted.

This paper would benefit from a table of symbols/notation, or at a minimum more clear definitions of key notation. I can make sense of it based on familiarity with related literature, but I do not think the equations stand on their own. For example, I can guess about implicit outer products in Equation 2, but considering that $cc$ was used two lines above to refer to compression in the $c$ axis direction rather than an outer product, it is really confusing here. In another, inconsistency in use of $h$ and $f$ in equations 7, 11, 12 leaves me confused—I cannot tell if this is two names for the same thing or some notation that I do not understand. I did not understand what $\boldsymbol{n}$ is in Equation 21 and 22 until much later. A table would help all of these.

There are times that the paper is insufficiently specific about the category of models considered, which leads to statements that are incorrect at the generality in which they are presented.

Microstructural models do some things that this paper suggests are impossible (e.g., model dislocation density, L136). In general, microstructural models like ELLE (Llorens et al., 2022) deserve mention in the introduction, as potentially the best available tool to consider fabric development in 0D. A close read by the authors, making sure that all categorical statements apply to all models, or that statements are qualified, is needed.

The paper is a bit under-referenced. It gets better as it goes on, so this is most evident in the introduction; several claims need citation, and there are several places where additional references were missed. Each of the first three sentences in the introduction deserve citations. The list of studies at lines 32 to 33 misses several recent references that used coupled flow/fabric modeling: (Gillet-Chaulet et al., 2006) considered an idealized divide/flank; (Rathmann and Lilien, 2022a) considered an idealized ice stream. (Lilien et al., 2023) was coupled used real geometry on a flowline at a divide and (Gerber et al., 2023) was also coupled used real geometry in an ice stream. The claim that coupled modeling studies can be counted on one hand needs to be amended. Similarly in lines 30-36, it is not really fair to skip the ESTAR line of work (Graham et al., 2018; McCormack et al., 2022), which lies intermediate to the two approaches mentioned and has been used for relatively large scale simulation of Thwaites. At line 43, the authors are missing a number of direct measurements in faster-flowing areas, such as (Jackson and Kamb, 1997; Voigt, 2017) among a handful of others. I think the current reference for the EGRIP fabric is (Stoll et al., 2024), though this may not have been out at the time that this preprint was submitted. At line 54, (Lilien et al., 2023) has a similar conclusion to (Richards et al., 2023). At line 57, I would argue that (Martín et al., 2009; Pettit et al., 2007) are both anisotropic rheologies in their own right. Certainly both (Rathmann and Lilien, 2022a, b) have different nonlinear rheologies that they use, so the list is an undercount of proliferation. The temperature in line 396 needs a citation.

Title: This study does not really compare fabric evolution models, but rather unites them.
L56: This claim is not really correct. We have long had laboratory deformation tests, so it seems untrue that it has not been possible to test anisotropic rheologies.
L125: Only appears true at depth in ice sheets, whereas near the surface NGG is normally conceptualized as something like a part of firnification.
L140: is $c_i$ really scalar?
L226: There are infinite possible assumptions, and indeed others have been used (like the linear combination of these)
L291: from not form
L429: Why not include the Martin approximation from Rathmann and Lilien, too?
L437: Latex error in min
L438: extra "two"
L467: Need subscript on the strain rate

**References**

Gerber, T. A., Lilien, D. A., Rathmann, N. M., Franke, S., Young, T. J., Valero-Delgado, F., Ershadi, M. R., Drews, R., Zeising, O., Humbert, A., Stoll, N., Weikusat, I., Grinsted, A., Hvidberg, C. S., Jansen, D., Miller, H., Helm, V., Steinhage, D., O'Neill, C., Paden, J., Gogineni, S. P., Dahl-Jensen, D., and Eisen, O.: Crystal orientation fabric anisotropy causes directional

hardening of the Northeast Greenland Ice Stream, Nat Commun, 14, 2653, https://doi.org/10.1038/s41467-023-38139-8, 2023.

Gillet-Chaulet, F., Gagliardini, O., Meyssonnier, J., Zwinger, T., and Ruokolainen, J.: Flow-induced anisotropy in polar ice and related ice-sheet flow modelling, J. Non-Newtonian Fluid Mech, 134, 33–43, https://doi.org/10.1016/j.jnnfm.2005.11.005, 2006.

Graham, F. S., Morlighem, M., Warner, R. C., and Treverrow, A.: Implementing an empirical scalar constitutive relation for ice with flow-induced polycrystalline anisotropy in large-scale ice sheet models, The Cryosphere, 12, 1047–1067, https://doi.org/10.5194/tc-12-1047-2018, 2018.

Jackson, M. and Kamb, B.: The marginal shear stress of Ice Stream B, West Antarctica, Journal of Glaciology, 43, 415–426, https://doi.org/10.3189/S0022143000035000, 1997.

Lilien, D. A., Rathmann, N. M., Hvidberg, C. S., Grinsted, A., Ershadi, M. R., Drews, R., and Dahl-Jensen, D.: Simulating higher-order fabric structure in a coupled, anisotropic ice-flow model: application to Dome C, Journal of Glaciology, 1–20, https://doi.org/10.1017/jog.2023.78, 2023.

Llorens, M.-G., Griera, A., Bons, P. D., Weikusat, I., Prior, D. J., Gomez-Rivas, E., de Riese, T., Jimenez-Munt, I., García-Castellanos, D., and Lebensohn, R. A.: Can changes in deformation regimes be inferred from crystallographic preferred orientations in polar ice?, The Cryosphere, 16, 2009–2024, https://doi.org/10.5194/tc-16-2009-2022, 2022.

Martín, C., Gudmundsson, G. H., Pritchard, H. D., and Gagliardini, O.: On the effects of anisotropic rheology on ice flow, internal structure, and the age-depth relationship at ice divides, Journal of Geophysical Research, 114, F04001, https://doi.org/10.1029/2008JF001204, 2009.

McCormack, F. S., Warner, R. C., Seroussi, H., Dow, C. F., Roberts, J. L., and Treverrow, A.: Modeling the Deformation Regime of Thwaites Glacier, West Antarctica, Using a Simple Flow Relation for Ice Anisotropy (ESTAR), Journal of Geophysical Research: Earth Surface, 127, e2021JF006332, https://doi.org/10.1029/2021JF006332, 2022.

Pettit, E. C., Thorsteinsson, T., Jacobson, H. P., and Waddington, E. D.: The role of crystal fabric in flow near an ice divide, Journal of Glaciology, 53, 277–288, https://doi.org/10.3189/172756507782202766, 2007.

Rathmann, N. M. and Lilien, D. A.: Inferred basal friction and mass flux affected by crystal-orientation fabrics, Journal of Glaciology, 68, 236–252, https://doi.org/10.1017/jog.2021.88, 2022a.

Rathmann, N. M. and Lilien, D. A.: On the nonlinear viscosity of the orthotropic bulk rheology, Journal of Glaciology, 68, 1243–1248, https://doi.org/10.1017/jog.2022.33, 2022b.

Richards, D. H., Pegler, S. S., Piazolo, S., Stoll, N., and Weikusat, I.: Bridging the Gap Between Experimental and Natural Fabrics: Modeling Ice Stream Fabric Evolution and its Comparison With Ice-Core Data, Journal of Geophysical Research: Solid Earth, 128, e2023JB027245, https://doi.org/10.1029/2023JB027245, 2023.

Stoll, N., Weikusat, I., Jansen, D., Bons, P., Darányi, K., Westhoff, J., Llorens, M.-G., Wallis, D., Eichler, J., Saruya, T., Homma, T., Drury, M., Wilhelms, F., Kipfstuhl, S., Dahl-Jensen, D., and Kerch, J.: EastGRIP ice core reveals the exceptional evolution of crystallographic preferred orientation throughout the Northeast Greenland Ice Stream, EGUsphere, 1–34, https://doi.org/10.5194/egusphere-2024-2653, 2024.

Voigt, D. E.: c-Axis Fabric of the South Pole Ice Core, SPC14, https://doi.org/10.15784/601057, 2017.

---

## Author Comment (AC2)

**Review 1**

This paper presents a unified framework in which most recent, large-scale ice-crystal fabric modeling can be considered. The challenge in constructing the framework was mainly in the non-collinearity between stress and strain rate with anisotropic rheologies, which bleeds into the fabric evolution. The unified framework clarifies this effect of anisotropy and allows for using quite a few different rheologies in a unified model of fabric evolution. This unified fabric model is used to try to reproduce data from the GRIP and EGRIP ice cores, with 7 different rheologies, and parameters tuned for each rheology to fit observations. Several of the rheologies allow the model to do a good job reproducing the fabric observations. From these results, the authors recommend one of these rheologies for use going forward.

**General comments**

This is an interesting study, and I enjoyed reading it. The writing is good though occasionally repetitive, and the figures are well made. While this framework is not a huge change from the existing descriptions of lattice rotation, the use of the normalized stress to support this general framework is a nice addition, and the framework provides clarity on how parameters relate between models. Overall, I think it is valuable to be able to compare these rheologies in a common framework for fabric evolution, and I am quite impressed by the number of rheologies compared.

We thank the reviewer for their positive and detailed comments, and we believe that by addressing the points they mention the manuscript will be a stronger paper. Please find our detailed responses below

However, I think the authors should temper and reframe the latter part of work; this is a convoluted way to test rheologies, and I am not convinced that it should be used for this purpose. Contrary to the claim in the introduction, it has in fact been possible to test rheologies—we have laboratory deformation tests against which to compare rheologies, and this has been done by one of the studies considered extensively within the present manuscript (Rathmann and Lilien, 2022b). Comparing to deformation tests is a much, much more direct test of the rheologies than feeding them through a fabric evolution with tuned parameters, uncertain flow history, and simple assumptions about flow (shallow shelf or Dansgaard-Johnsen). As the authors point out late in this work, the Sach's hypothesis cannot produce the extent of enhancement observed with real fabrics by about an order of magnitude—that is a lot more straightforward than sending it through a fabric model! This work tuned multiple parameters, one of which carries physical meaning (lambda), to make these rheologies reproduce observed fabrics—without knowing that the value of lambda is reasonable, how do we know that a rheology gets the right fabric for the right reason, rather than due to this tuning? I do find the comparison of fabric predicted using Taylor, Estar, CAFFE illuminating, since I would have expected them to do better, but I am not convinced that this is a meaningful way to distinguish between the other rheologies considering the tuning and uncertainty. As a result, I would like to see the conclusions tempered. In terms of rheologies, I think all I conclude is that most of them can be tuned to do a pretty good job reproducing observations of fabric—not that one of Sachs, Rathmann, or GOLF is superior based on the present work. I think the work is publishable in The Cryosphere with relatively minor changes if this rheology recommendation is tempered/removed.

We are happy to temper the conclusions as suggested, indeed the modelling in this paper is only looking at two single locations in the whole ice sheet. We also acknowledge that data from laboratory experiments exists. However, based on our previous work (Richards et al. 2023), we found that the fabric evolution model that worked well in laboratory experiments (Richards et al. 2021) did not work when applied to the natural world. This result led us to the hypothesis that, at the lower strain rates of the natural world (about 5 orders of magnitude), different physical processes are dominant at the micro-scale. This hypothesis motivated this work; in which we try to compare rheologies at the natural strain-rate scale. We will emphasise this idea more in the introduction and highlight the advantages and disadvantages of this method compared to using laboratory data, as well as tempering the rheology recommendation as suggested.

**Specific comments**

First, it is Dr. Pettit, not Petit.

We thank the reviewer for pointing out this mistake and will correct it.

I think there are implicit assumptions about collinearity of stress and strain that might confuse the reader. At a minimum, the assumptions should be stated. Equation 6 does not appear to be a result of Equations 2 and 5, and I think it involves more assumptions that the authors are crediting. Clearly plugging equation 2 into 5 does not straightforwardly lead to 6—perhaps I am missing a complicated derivation demonstrating this, but I do not think that I am. Rather, I think that this is essentially an assumption about the collinearity of the strain rate and stress. If I understand, 6 is essentially an alternative to 5, and then by combining them we get something like the fabric evolution equation in 44? Considering the conclusions of the paper, this does not really make sense—the reader needs a full accounting of whether Eq. 6 is alternative to the strain-rate based model (as I read it) or an equivalent formulation as currently implied. Relatedly, I think the introduction of normalized stress is muddled, which might confuse the reader about what is colinear under that assumption. In essence, I think Eq. 28 is essentially a property of 29 that emerges for isotropy, but it took me a long time to see this. If we had a clear definition of normalized stress early, that was then consistent across rheologies, I would have an easier time conceptualizing the results.

Eq (6) does in fact come from directly plugging Eq (2) into Eq(5). We will include this derivation in the supplement as it is not straightforward. In index notation, Eq (2) and Eq (5) are:

$$\dot{\varepsilon}'_{ij} = \eta^{-1}\left(\tau'_{ij} + \frac{3(E_{cc}-1)-4(E_{ca}-1)}{2}\tau'_{kl}c_k c_l c_i c_j + (E_{ca}-1)(\tau'_{ik}c_k c_j + c_i c_k \tau'_{kj}) - \frac{E_{cc}-1}{2}\tau'_{kl}c_k c_l \delta_{ij}\right) \tag{R1}$$

$$\frac{dc_i}{dt} = \omega'_{ij}c_j - (\dot{\varepsilon}'_{ij}c_j - \dot{\varepsilon}'_{kj}c_k c_j c_i), \tag{R2}$$

Taking advantage of the fact that $c_i c_i = 1$, and defining $\tau'_c = \tau'_{ij}c_j c_i = \tau'_{ji}c_j c_i = \tau'_{kl}c_k c_l$ etc.

$$\dot{\varepsilon}'_{ij}c_j = \eta^{-1}\left(\tau'_{ij}c_j + \frac{3(E_{cc}-1)-4(E_{ca}-1)}{2}\tau'_c c_i + (E_{ca}-1)(\tau'_{ik}c_k + c_i \tau'_c) - \frac{E_{cc}-1}{2}\tau'_c c_i\right) \tag{R3}$$

and

$$\dot{\varepsilon}'_{ij}c_i c_j = \eta^{-1}\left(\tau'_c + \frac{3(E_{cc}-1)-4(E_{ca}-1)}{2}\tau'_c + (E_{ca}-1)(\tau'_c + \tau'_c) - \frac{E_{cc}-1}{2}\tau'_c\right) \tag{R4}$$

hence

$$\dot{\varepsilon}'_{kj}c_k c_j c_i = \eta^{-1}\left(\tau'_c c_i + \frac{3(E_{cc}-1)-4(E_{ca}-1)}{2}\tau'_c c_i + (E_{ca}-1)(\tau'_c c_i + \tau'_c c_i) - \frac{E_{cc}-1}{2}\tau'_c c_i\right) \tag{R5}$$

Then, subtracting Eq. (R5) from (R3), and renaming indices in Eq. (R3) so that $\tau'_{ik}c_k \rightarrow \tau'_{ij}c_j$:

$$\dot{\varepsilon}'_{ij}c_j - \dot{\varepsilon}'_{kj}c_k c_j c_i = \eta^{-1}\left((\tau'_{ij}c_j - \tau'_c c_i) + (E_{ca}-1)(\tau'_{ij}c_j - \tau'_c c_i)\right) \tag{R6}$$

finally giving:

$$\dot{\varepsilon}'_{ij}c_j - \dot{\varepsilon}'_{kj}c_k c_j c_i = \eta^{-1}E_{ca}(\tau'_{ij}c_j - \tau'_{kj}c_k c_j c_i) \tag{R7}$$

Consequently, this does not require any assumptions about co-linearity of the stress and strain-rate.

Regarding the definition of the normalised stress, the reviewer is correct that Eq (28) is a property of Eq (29) under isotropy, and we agree that the normalised stress could be defined in a clearer manner, and we will define it earlier as the reviewer suggests.

It does not really make sense to try to fit these rheologies to data when they are mis-constrained at the surface. That is to say, it appears that the models struggle at EGRIP in part because the shallowest available data are not that close to isotropy. I suggest assuming that reorientation processes differ in the firn, justifying the use of the shallowest measurement of fabric as the initial state for the model. In this way, the misfit at the top of the ice sheet, which I do not think is due to the models per se, can be avoided, and they can be compared over more depths.

[Figure]

Figure 1: Evolution of eigenvalues at EGRIP with depth with Sachs model, for different initial conditions

We agree with the reviewer that the fabrics at the shallowest depth are due to firn processes which we are not modelling. However, we find that the fabric is insensitive to the initial condition below a depth of 400, as shown in Fig. . In the manuscript, when fitting the parameters, we only include data below this 400 m as part of the inversion.

I do not see why the authors exclude migration recrystallization under the Taylor hypothesis—including it by using the strain rate to calculate s' seems like a reasonable alternative to allow better comparison, and presumably would be very easy to do. Particularly since it doesn't need to be coded up, as beta is taken to be zero, this is no work to include.

Based on our third comment, where we establish that we are not making any assumptions in Eq. (6) but rather this follows from the above derivation, we are not making a Sachs or Taylor assumption in our derivation of migration recrystallization in section 2.3.1. We then set all migration recrystallization to zero for the rest of the paper.

On the topic of migration recrystallization, there should be discussion on how the possibility of low temperature, high stress/strain migration recrystallization might affect the results. There is some work that suggests such a possibility (see the Faria 2014 reviews), so its effect on the results here should be considered. I do not necessarily see a need to tune Beta, but discussion is warranted.

We are happy to add this into the discussion, we thank the reviewer for highlighting this.

This paper would benefit from a table of symbols/notation, or at a minimum more clear definitions of key notation. I can make sense of it based on familiarity with related literature, but I do not think the equations stand on their own. For example, I can guess about implicit outer products in Equation 2, but considering that cc was used two lines above to refer to compression in the c axis direction rather than an outer product, it is really confusing here. In another, inconsistency in use of h and f in equations 7, 11, 12 leaves me confused—I cannot tell if this is two names for the same thing or some notation that I do not understand. I did not understand what n is in Equation 21 and 22 until much later. A table would help all of these.

We will add a table of symbols and mathematical operations to aid the reader. We have aimed to be consistent throughout the paper but agree with the the reviewer that the examples highlighted are

*confusing, and will change them. In particular, we can update Eq. (1) to not result in cc having two different meanings.*

> There are times that the paper is insufficiently specific about the category of models considered, which leads to statements that are incorrect at the generality in which they are presented.

> Microstructural models do some things that this paper suggests are impossible (e.g., model dislocation density, L136). In general, microstructural models like ELLE (Llorens et al., 2022) deserve mention in the introduction, as potentially the best available tool to consider fabric development in 0D. A close read by the authors, making sure that all categorical statements apply to all models, or that statements are qualified, is needed.

*We thank the reviewer for highlighting this and will add this into the introduction and update the paper to ensure all these statements are correct.*

> The paper is a bit under-referenced. It gets better as it goes on, so this is most evident in the introduction; several claims need citation, and there are several places where additional references were missed. Each of the first three sentences in the introduction deserve citations. The list of studies at lines 32 to 33 misses several recent references that used coupled flow/fabric modeling: (Gillet-Chaulet et al., 2006) considered an idealized divide/flank; (Rathmann and Lilien, 2022a) considered an idealized ice stream. (Lilien et al., 2023) was coupled used real geometry on a flowline at a divide and (Gerber et al., 2023) was also coupled used real geometry in an ice stream. The claim that coupled modeling studies can be counted on one hand needs to be amended. Similarly in lines 30-36, it is not really fair to skip the ESTAR line of work (Graham et al., 2018; McCormack et al., 2022), which lies intermediate to the two approaches mentioned and has been used for relatively large scale simulation of Thwaites. At line 43, the authors are missing a number of direct measurements in faster-flowing areas, such as (Jackson and Kamb, 1997; Voigt, 2017) among a handful of others. I think the current reference for the EGRIP fabric is (Stoll et al., 2024), though this may not have been out at the time that this preprint was submitted. At line 54, (Lilien et al., 2023) has a similar conclusion to (Richards et al., 2023). At line 57, I would argue that (Martín et al., 2009; Pettit et al., 2007) are both anisotropic rheologies in their own right. Certainly both (Rathmann and Lilien, 2022a, b) have different nonlinear rheologies that they use, so the list is an undercount of proliferation. The temperature in line 396 needs a citation.

*We thank the reviewer for highlighting this and suggesting these references and will add these into the paper and make sure the introduction is sufficiently referenced.*

> Title: This study does not really compare fabric evolution models, but rather unites them.

*We are happy to change this to something like "Unifying and testing fabric evolution models and anisotropic rheologies"*

> L56: This claim is not really correct. We have long had laboratory deformation tests, so it seems untrue that it has not been possible to test anisotropic rheologies.

*We are happy to update this to "Secondly, there has been no way to test whether an anisotropic rheology at strain-rates seen in ice sheet", as laboratory experiments are at much higher strain rates. Laboratory experiments also cannot measure all of the the different components of the anisotropic stress tensor which is needed to characterise anisotropic rheologies.*

> L125: Only appears true at depth in ice sheets, whereas near the surface NGG is normally conceptualized as something like a part of firnification.

*We thank the reviewer for this clarification and will update the text with references*

> L140: is $c_i$ really scalar?

*This section of the text has not been updated from a previous iteration using index notation, and we thank the reviwer for spotting this and will update it to $\boldsymbol{c}$*

> L226: There are infinite possible assumptions, and indeed others have been used (like the linear combination of these)

We will re-word the text to say end members rather than possible assumptions

L291: from not form

We thank the reviewer for spotting this

L429: Why not include the Martin approximation from Rathmann and Lilien, too?

We are happy to add this. We have tested it and it gives very similar results to the unapproximated version

L437: Latex error in min

L438: extra "two"

L467: Need subscript on the strain rate

We thank the reviewer for finding these errors and will correct them

---

## Author Response (AR1)

**Review 1**

This paper presents a unified framework in which most recent, large-scale ice-crystal fabric modeling can be considered. The challenge in constructing the framework was mainly in the non-collinearity between stress and strain rate with anisotropic rheologies, which bleeds into the fabric evolution. The unified framework clarifies this effect of anisotropy and allows for using quite a few different rheologies in a unified model of fabric evolution. This unified fabric model is used to try to reproduce data from the GRIP and EGRIP ice cores, with 7 different rheologies, and parameters tuned for each rheology to fit observations. Several of the rheologies allow the model to do a good job reproducing the fabric observations. From these results, the authors recommend one of these rheologies for use going forward.

**General comments**

This is an interesting study, and I enjoyed reading it. The writing is good though occasionally repetitive, and the figures are well made. While this framework is not a huge change from the existing descriptions of lattice rotation, the use of the normalized stress to support this general framework is a nice addition, and the framework provides clarity on how parameters relate between models. Overall, I think it is valuable to be able to compare these rheologies in a common framework for fabric evolution, and I am quite impressed by the number of rheologies compared.

We thank the reviewer for their positive and detailed comments, and we believe that by addressing the points they mention the manuscript will be a stronger paper. Please find our detailed responses below. The reviewers comments are in black and our responses are in blue.

However, I think the authors should temper and reframe the latter part of work; this is a convoluted way to test rheologies, and I am not convinced that it should be used for this purpose. Contrary to the claim in the introduction, it has in fact been possible to test rheologies—we have laboratory deformation tests against which to compare rheologies, and this has been done by one of the studies considered extensively within the present manuscript (Rathmann and Lilien, 2022b). Comparing to deformation tests is a much, much more direct test of the rheologies than feeding them through a fabric evolution with tuned parameters, uncertain flow history, and simple assumptions about flow (shallow shelf or Dansgaard-Johnsen). As the authors point out late in this work, the Sach's hypothesis cannot produce the extent of enhancement observed with real fabrics by about an order of magnitude—that is a lot more straightforward than sending it through a fabric model! This work tuned multiple parameters, one of which carries physical meaning (lambda), to make these rheologies reproduce observed fabrics—without knowing that the value of lambda is reasonable, how do we know that a rheology gets the right fabric for the right reason, rather than due to this tuning? I do find the comparison of fabric predicted using Taylor, Estar, CAFFE illuminating, since I would have expected them to do better, but I am not convinced that this is a meaningful way to distinguish between the other rheologies considering the tuning and uncertainty. As a result, I would like to see the conclusions tempered. In terms of rheologies, I think all I conclude is that most of them can be tuned to do a pretty good job reproducing observations of fabric—not that one of Sachs, Rathmann, or GOLF is superior based on the present work. I think the work is publishable in The Cryosphere with relatively minor changes if this rheology recommendation is tempered/removed.

1. We thank the reviewer for this detailed comment, however we do disagree about the relevance of this approach vs laboratory experiments. Motivating this work, we believe there are two issues with laboratory experiments: firstly, that the experiments are done at much higher strain rates/ stresses than what is seen at locations like EGRIP and GRIP. In our previous work [Richards et al., 2023], we found that the fabric evolution model that worked well in laboratory experiments [Richards et al., 2021] did not work when applied to the natural world. We have added the following paragraph to explain our reasoning in the introduction:

A key difficulty in modelling fabrics and their effect arises from our lack of understanding of the underlying physics. A large range of fabric measurements are available from deformation tests in the laboratory, extending from the present day to the 1950s [Glen, 1952, Craw et al.,

2018, Journaux et al., 2019]. Yet experiments need to take place at strain rates around 5 orders of magnitude greater than those seen in the natural world to allow for experiments that last for reasonable durations (days to weeks). Some of our recent work found that models for fabric evolution calibrated against laboratory experiments [Richards et al., 2021] could not reproduce fabrics observed in the natural world [Richards et al., 2023], suggesting that different microstructural processes are occurring at the lower strain rates observed in ice sheets. This partly motivates this present work, where we seek to constrain models of anisotropy directly against observations in the natural world.

Secondly, it is difficult with laboratory experiments to measure the different components of the anisotropic viscosity tensor. For example, in a unconfined compression test under constant strain rate, only the vertical stress is measured, giving only 1 out of the many components of the anisotropic tensor. We have tried to explain this with the following sentences slightly later in the introduction:

> Secondly, there has been no way to test whether an anisotropic rheology is accurately representing the anisotropic effect. Unlike an isotropic material, ice with a fabric will have a different mechanical response when deformed in different directions, and stresses in one direction can induce a range of deformations in other directions (what we refer to as the anisotropic effect). A laboratory experiment measures stress and deformation in the same direction giving only 1 out of the 36 components of the anisotropic viscosity tensor. This lack of information has led to a proliferation of competing rheologies [Gillet-Chaulet et al., 2005, Pettit et al., 2007, Martín et al., 2009, Placidi et al., 2010, Graham et al., 2018, Rathmann and Lilien, 2021, 2022], with no consensus on which may be most appropriate.

We do agree that this method carries much more noise and uncertainty than laboratory experiments, but we do believe that it provides insights into a) low strain rates and b) the anisotropic effect that experiments cannot. The reviewer correctly points out that this method does contain limitations, and based on this we have slightly tempered the conclusions, in particular we have removed 'We recommend the Rathmann rheology' at the start of section 7.3. We also hope the reviewer agrees that, in light of our reply to the comment below where we clarify that the model does not require any assumptions about the co-linearity of the stress and strain rate, our conclusions are more mathematically justified than may have initially appeared.

Our recommendation of the Rathmann rheology is based on the fact that of the anisotropic rheologies (Rathmann, GOLF, Sachs), the GOLF rheology predicts the correct magnitude but is not capable of predicting a girdle pattern at EGRIP no matter what parameters are chosen. The Sachs model predicts the fabric well but as you say, underpredicts the magnitude of the anisotropic effect, leaving the Rathmann rheology as the only one that does both.

**Specific comments**

> First, it is Dr. Pettit, not Petit.

2. We thank the reviewer for pointing out this mistake and will correct it.

> I think there are implicit assumptions about collinearity of stress and strain that might confuse the reader. At a minimum, the assumptions should be stated. Equation 6 does not appear to be a result of Equations 2 and 5, and I think it involves more assumptions that the authors are crediting. Clearly plugging equation 2 into 5 does not straightforwardly lead to 6—perhaps I am missing a complicated derivation demonstrating this, but I do not think that I am. Rather, I think that this is essentially an assumption about the collinearity of the strain rate and stress. If I understand, 6 is essentially an alternative to 5, and then by combining them we get something like the fabric evolution equation in 44? Considering the conclusions of the paper, this does not really make sense—the reader needs a full accounting of whether Eq. 6 is alternative to the strain-rate based model (as I read it) or an equivalent formulation as currently implied.

3. Eq (6) does in fact come from directly plugging Eq (2) into Eq(5). In index notation, Eq (2) and Eq (5) are:

$$\dot{\varepsilon}'_{ij} = \eta^{-1}\left(\tau'_{ij} + \frac{3(E_{cc}-1)-4(E_{ca}-1)}{2}\tau'_{kl}c_k c_l c_i c_j + (E_{ca}-1)(\tau'_{ik}c_k c_j + c_i c_k \tau'_{kj}) - \frac{E_{cc}-1}{2}\tau'_{kl}c_k c_l \delta_{ij}\right)$$

$$\text{(R1)}$$

$$\frac{dc_i}{dt} = \omega'_{ij}c_j - (\dot{\varepsilon}'_{ij}c_j - \dot{\varepsilon}'_{kj}c_k c_j c_i), \tag{R2}$$

Taking advantage of the fact that $c_i c_i = 1$, and defining $\tau'_c = \tau'_{ij}c_j c_i = \tau'_{ji}c_j c_i = \tau'_{kl}c_k c_l$ etc.

$$\dot{\varepsilon}'_{ij}c_j = \eta^{-1}\left(\tau'_{ij}c_j + \frac{3(E_{cc}-1)-4(E_{ca}-1)}{2}\tau'_c c_i + (E_{ca}-1)(\tau'_{ik}c_k + c_i \tau'_c) - \frac{E_{cc}-1}{2}\tau'_c c_i\right) \tag{R3}$$

and

$$\dot{\varepsilon}'_{ij}c_i c_j = \eta^{-1}\left(\tau'_c + \frac{3(E_{cc}-1)-4(E_{ca}-1)}{2}\tau'_c + (E_{ca}-1)(\tau'_c + \tau'_c) - \frac{E_{cc}-1}{2}\tau'_c\right) \tag{R4}$$

hence

$$\dot{\varepsilon}'_{kj}c_k c_j c_i = \eta^{-1}\left(\tau'_c c_i + \frac{3(E_{cc}-1)-4(E_{ca}-1)}{2}\tau'_c c_i + (E_{ca}-1)(\tau'_c c_i + \tau'_c c_i) - \frac{E_{cc}-1}{2}\tau'_c c_i\right) \tag{R5}$$

Then, subtracting Eq. (R5) from (R3), and renaming indices in Eq. (R3) so that $\tau'_{ik}c_k \to \tau'_{ij}c_j$:

$$\dot{\varepsilon}'_{ij}c_j - \dot{\varepsilon}'_{kj}c_k c_j c_i = \eta^{-1}\left((\tau'_{ij}c_j - \tau'_c c_i) + (E_{ca}-1)(\tau'_{ij}c_j - \tau'_c c_i)\right) \tag{R6}$$

finally giving:

$$\dot{\varepsilon}'_{ij}c_j - \dot{\varepsilon}'_{kj}c_k c_j c_i = \eta^{-1}E_{ca}(\tau'_{ij}c_j - \tau'_{kj}c_k c_j c_i) \tag{R7}$$

Consequently, this does not require any assumptions about co-linearity of the stress and strain-rate. We have added this derivation as an appendix and signposted it

> Relatedly, I think the introduction of normalized stress is muddled, which might confuse the reader about what is colinear under that assumption. In essence, I think Eq. 28 is essentially a property of 29 that emerges for isotropy, but it took me a long time to see this. If we had a clear definition of normalized stress early, that was then consistent across rheologies, I would have an easier time conceptualizing the results.

4. Regarding the definition of the normalised stress, the reviewer is correct that Eq (28) is a property of Eq (29) under isotropy. We have re-done this section in the text, first defining the isotropic response of the Sachs rheology and then defining the normalised stress directly in terms of the original stress.

> It does not really make sense to try to fit these rheologies to data when they are mis-constrained at the surface. That is to say, it appears that the models struggle at EGRIP in part because the shallowest available data are not that close to isotropy. I suggest assuming that reorientation processes differ in the firn, justifying the use of the shallowest measurement of fabric as the initial state for the model. In this way, the misfit at the top of the ice sheet, which I do not think is due to the models per se, can be avoided, and they can be compared over more depths.

5. We agree with the reviewer that the fabrics at the shallowest depth are due to firn processes which we are not modelling. However, we find that the fabric is insensitive to the initial condition below a depth of 400, as shown in the figure below. In the manuscript, when fitting the parameters, we only include data below this 400 m as part of the inversion. We have added this figure to the supplement and added the following (in italics) to the main text:

> 'We do not include the effect of firn in our analysis, which significantly affects the observed fabric above a depth of around 550m. Hence, we only compare the models to observations below a depth of 550 m. *Although different initial conditions for the fabric could perhaps fit the observations better, we find that the results are insensitive to initial condition below a depth of 400 m (Fig S13).*

[Figure]

Figure 1: Evolution of eigenvalues at EGRIP with depth with Sachs model, for different initial conditions

I do not see why the authors exclude migration recrystallization under the Taylor hypothesis—including it by using the strain rate to calculate s' seems like a reasonable alternative to allow better comparison, and presumably would be very easy to do. Particularly since it doesn't need to be coded up, as beta is taken to be zero, this is no work to include.

6. Based on comment 3, where we establish that we are not making any assumptions in Eq. (6) but rather this follows from the above derivation, we are not making a Sachs or Taylor assumption in our derivation of migration recrystallization in section 2.3.1. We then set all migration recrystallization to zero for the rest of the paper.

On the topic of migration recrystallization, there should be discussion on how the possibility of low temperature, high stress/strain migration recrystallization might affect the results. There is some work that suggests such a possibility (see the Faria 2014 reviews), so its effect on the results here should be considered. I do not necessarily see a need to tune Beta, but discussion is warranted.

6. We have added the following in the parameter estimation section (new text in italics):

The GRIP ice divide (above a depth of 2500 m) and the EGRIP ice stream are at $T < -20°$ C *and strain rates $< 10^{-11}$ $s^{-1}$.* Migration recrystallization is known to be predominantly a high temperature [Craw et al., 2018, Qi et al., 2019] or *potentially also a low temperature high strain rate phenomena [Faria et al., 2014]. As both GRIP and EGRIP are at low temperature and low strain rates (relative to laboratory strain rates), we do not think migration recrystallization will be significant for either location and set it to zero.*

This paper would benefit from a table of symbols/notation, or at a minimum more clear definitions of key notation. I can make sense of it based on familiarity with related literature, but I do not think the equations stand on their own. For example, I can guess about implicit outer products in Equation 2, but considering that cc was used two lines above to refer to compression in the c axis direction rather than an outer product, it is really confusing here. In another, inconsistency in use of h and f in equations 7, 11, 12 leaves me confused—I cannot tell if this is two names for the same thing or some notation that I do not understand. I did not understand what n is in Equation 21 and 22 until much later. A table would help all of these.

7. We thank the reviewer for pointing this out. In light of this we have added a table of symbols. We have also decided to switch all the equations containing tensor algebra into index notation, in order to make it easier to follow for readers not familiar with the literature. We hope this clarifies the confusion the reviewer noted regarding $E_{cc}$ and the following equation. Furthermore, we have modified Eqs. 7, 11, 12 to clarify we a referring to some arbitrary function $f$, for which the definition is not yet important. We have also added text after Eq. (21) to clarify it's meaning:

> here $\boldsymbol{n}$ is the space of all possible orientations a single $\boldsymbol{c}$-axis may have, and $\boldsymbol{v}$ and $s'$ are functions over $\boldsymbol{n}$ and $\nabla^*$ is the gradient operator over $\boldsymbol{n}$ (not physical space $\boldsymbol{x}$) restricted to the space of possible orientations:

> There are times that the paper is insufficiently specific about the category of models considered, which leads to statements that are incorrect at the generality in which they are presented. Microstructural models do some things that this paper suggests are impossible (e.g., model dislocation density, L136). In general, microstructural models like ELLE (Llorens et al., 2022) deserve mention in the introduction, as potentially the best available tool to consider fabric development in 0D. A close read by the authors, making sure that all categorical statements apply to all models, or that statements are qualified, is needed.

8. We thank the reviewer for highlighting this and have added the following paragraph to the introduction:

> At the other end of the scale, there exist models which directly model grain dynamics or dislocation dynamics directly [see Montagnat et al., 2014, for a review]. These models can provide valuable insight into the underlying physics and some have been applied to model fabrics in ice sheets [Llorens et al., 2022]. However they are too numerically expensive to be coupled to a large-scale ice flow model. In this contribution we focus solely on fabric evolution models which are computationally inexpensive enough to be included in large-scale models, by making assumptions about grain dynamics rather than simulating them directly.

We have also added in L136 (new text in italics):

> Dislocations exist at the molecular scale so *for any fabric evolution model aiming to be applied at the large scale they* cannot be modelled directly

> The paper is a bit under-referenced. It gets better as it goes on, so this is most evident in the introduction; several claims need citation, and there are several places where additional references were missed. Each of the first three sentences in the introduction deserve citations. The list of studies at lines 32 to 33 misses several recent references that used coupled flow/fabric modeling: (Gillet-Chaulet et al., 2006) considered an idealized divide/flank; (Rathmann and Lilien, 2022a) considered an idealized ice stream. (Lilien et al., 2023) was coupled used real geometry on a flowline at a divide and (Gerber et al., 2023) was also coupled used real geometry in an ice stream. The claim that coupled modeling studies can be counted on one hand needs to be amended. Similarly in lines 30-36, it is not really fair to skip the ESTAR line of work (Graham et al., 2018; McCormack et al., 2022), which lies intermediate to the two approaches mentioned and has been used for relatively large scale simulation of Thwaites. At line 43, the authors are missing a number of direct measurements in faster-flowing areas, such as (Jackson and Kamb, 1997; Voigt, 2017) among a handful of others. I think the current reference for the EGRIP fabric is (Stoll et al., 2024), though this may not have been out at the time that this preprint was submitted. At line 54, (Lilien et al., 2023) has a similar conclusion to (Richards et al., 2023). At line 57, I would argue that (Martín et al., 2009; Pettit et al., 2007) are both anisotropic rheologies in their own right. Certainly both (Rathmann and Lilien, 2022a, b) have different nonlinear rheologies that they use, so the list is an undercount of proliferation. The temperature in line 396 needs a citation.

9. We thank the reviewer for highlighting this and suggesting these references and have added all of them in, with the exception of Stoll et al. 2024 as it is still a preprint at the time of writing. We have cited the GRIP temperature record [Johnsen, 2003]

> Title: This study does not really compare fabric evolution models, but rather unites them.

10. We have changed this to: 'A unified framework for large-scale fabric evolution models and anisotropic rheologies'

L56: This claim is not really correct. We have long had laboratory deformation tests, so it seems untrue that it has not been possible to test anisotropic rheologies.

11. As mentioned in our reply no. 1, we believe there are some limitations with laboratory experiments, due to them being at higher than natural strain rates.

L125: Only appears true at depth in ice sheets, whereas near the surface NGG is normally conceptualized as something like a part of firnification.

12. We thank the reviewer for this clarification and have chnaged this, with new text in italics:

While grains *in the firn* or those undergoing no deformation may undergo normal grain growth *[Arthern et al., 2010]*, in an ice sheet which is always deforming, the primary process for grain growth will be migration recrystallization [Faria et al., 2014].

L140: is $c_i$ really scalar?

13. We thank the reviewer for spotting this and have updated it to $\boldsymbol{c}$

L226: There are infinite possible assumptions, and indeed others have been used (like the linear combination of these)

14. We have updated the text to say end members rather possible assumptions

L291: from not form

15. We thank the reviewer for spotting this and have corrected it

L429: Why not include the Martin approximation from Rathmann and Lilien, too?

16. We found this gave the same results as the unapproximated version, although we agree that there is no good reason to include the Pettit approximation and not the Martin approximation. Upon reflection, we have decided to remove the Rathmann rheology (now called NLO (Non-linear orthotropic) in the paper) with Pettit approximation from the results. We belive this still provides a relevant test of the rheologies while also removing clutter from the paper.

L437: Latex error in min

L438: extra "two"

L467: Need subscript on the strain rate

17. We thank the reviewer for finding these errors and have corrected them

**Review 2**

This paper offers a comprehensive and detailed study of fabric evolution models, both grain size and macroscopic scale, and apply the model to both an ice stream and an ice divide with a range of different anisotropic rheologies. It is quite interesting to see the gaps between these models and how they fit into the realistic world. Especially, the authors relate grain rotation and anisotropic rheology to their models and reproduce fabric observations at an ice stream, which is of great significance to current ice-stream studies, where the development of ice fabric and its influence on ice streams are often overlooked. Overall it is a well executed study, and I look forward to seeing it published. I believe this paper will be a useful reference for ice modellers and also draw wider attention from the ice community to the importance of ice fabric and anisotropy regarding ice flow.

We thank the reviewer for their positive comments and their taking the time to review the paper

I feel this is already solid work. I just have a few minor questions that I'd like to mention here.

1) Is there a reference for 'a factor of 8' in Line 34 for the viscosity enhancement factor.

We have changed this to *the effect of fabric is usually only represented by adding to the viscosity an enhancement factor, for which values between 3 to 8 have been suggested [Graham et al., 2018]*

Maybe it would also be good to move the statement like '(Line 75) For ice, slip along the basal plane - with unit normal denoted by the c-axis - is approximately 70 times easier than other slip systems (Duval et al., 1983).' earlier in the Introduction.

We would rather keep this statement here as although there is large enhancement observed for a single crystal, for a polycrystal the maximum enhancement observed is around 10 [Pimienta and Duval, 1987], so placing this factor of 70 near Line 34 would be drawing a false equivalence in our opinion.

2) Line 100 'all large-scale models have so far been limited to linear grains'. As for some large-scale models (e.g. Zhang et al., 2024), the problem is actually not due to linear grains but there are no real grains. Instead, ice particles simulate grain aggregates and each particle is defined with a power-law rheology.

We thank the reviewer for pointing this out and will update the text to highlight it is most, not all, models. - we have changed this to 'all large-scale models *which include grain dynamics* have so far been limited to linear grains'.

3) It seems the authors mainly consider the ice inside the ice stream but not at the ice stream's shear margins (such as Sections 5.2, 6.3). It would be good to mention this in the paper because the ice fabric at shear margins is different. And the power-law stress exponent $n$ would also be important regarding strong shearing areas. I understand normally it is fine when $n$ is considered as 3 in ice models (Line 451), but recent studies also suggest a higher $n = 4$ (Bons et al., 2018; Ranganathan and Minchew, 2024).

We agree with reviewer about the different fabrics in shear margins and will update the text to clarify we are referring to the flow inside an ice stream. We have added in Section 5.2: 'To predict fabrics at the EGRIP drill site, *located in the centre of an ice stream* and where ice fabric measurements are available'. Regarding $n = 4$, we agree it would be interesting to investigate but we believe is beyond the scope of this work, the results are generally unaffected by changing the value of $n$ - this can be seen in the parameter sensitivity part of the supplement.

4) The authors assume the fabric undergoes unconfined compression as it moves vertically down at the ice divide, and the deformation is steady over time and the observed fabric is simple - a single maximum. I was just wondering if the ice fabric would also rotate when considering Raymond bump effect (Martín et al., 2009) that ice layers are not flat in an extensional environment (e.g. Figure 3)?

Theoretically, exactly at the ice divide the deformation should be solely unconfined compression. In reality, an ice core will never be drilled exactly at the divide and the divide moves over time. Nevertheless, this deformation condition dominates and modelling ice divides this way is well established in the literature [Castelnau et al., 1996, Montagnat et al., 2012]. We have not modified the text here however, as we already say 'Firstly, it is assumed the core is drilled at a perfect ice divide, such that it only experiences unconfined compression'

Or I feel this question is also similar to point(3). At the beginning part of this paper (such as Abstract and Introduction), the authors say they can reproduce observations at both an ice divide and at an ice stream, but the locations and specific cases of GRIP and EGRIP are not mentioned. Perhaps it would be helpful to add a few words at least in the Introduction to specify the areas the authors want to focus on?

We agree with this point and have modified the text: We apply the fabric predictions from these rheologies to two specific locations *where ice cores have been drilled: an ice stream, from the aforementioned EGRIP site, and at an nearby ice divide from the earlier GRIP (Greenland Ice core Project).*

5) As for the last paragraph in Section 7.4 (Lines 662-666), I would stand that the ice stream at EGRIP is a recent formation and may not be as old as 16,000 years (e.g. Fig. 1 in Tabone et al., 2024). But as for the age of around 3000 years, I think it would not be a problem for the authors' results. Because the shear margins were established 2000 years ago from Jansen et al (2024), and before this stage, there could be convergent ice flow or proto ice stream without well-developed shear margins.

We thank the reviewer for pointing this and have added the following to the discussion 'Some recent work (Jansen et al. 2024) has suggested it may be as young as 2000 years, *although a region of convergent ice flow could have been present before this which would produce a similar fabric.*'

> The writing is good but some words or statements are repetitive and could be removed, such as,
>
> Line 14: 'approximately **the** the same stress'. One of 'the' can be deleted.
>
> Line 223: ' If we only have **have** macroscopic quantities'; ' in terms of these **these** macroscopic'
>
> Line 414: 'that the **the** final age '.

We thank the reviewer for noticing these repetitions and have corrected these, alongside further proof reading the manuscript

> Line 642-645: ' In this contribution, we have constrained a general model for fabric evolution against an ice divide and an ice stream... where previous models could not provide accurate predictions.' This part could be removed, as the authors have clearly stated this in other sections several times. There is no need to repeat it in the 'Limitations of this Analysis' section.

We agree with the reviewer and have removed this.

**References**

Robert J. Arthern, David G. Vaughan, Andrew M. Rankin, Robert Mulvaney, and Elizabeth R. Thomas. In situ measurements of Antarctic snow compaction compared with predictions of models. *Journal of Geophysical Research: Earth Surface*, 115(F3), September 2010. ISSN 0148-0227. doi: 10.1029/2009JF001306. URL https://doi.org/10.1029/2009JF001306. Publisher: John Wiley & Sons, Ltd.

O. Castelnau, Th. Thorsteinsson, J. Kipfstuhl, P. Duval, and G. R. Canova. Modelling fabric development along the GRIP ice core, central Greenland. *Annals of Glaciology*, 23:194–201, 1996. doi: 10.3189/S0260305500013446. Publisher: Cambridge University Press.

Lisa Craw, Chao Qi, David J. Prior, David L. Goldsby, and Daeyeong Kim. Mechanics and microstructure of deformed natural anisotropic ice. *Journal of Structural Geology*, 115:152–166, October 2018. ISSN 0191-8141. doi: 10.1016/j.jsg.2018.07.014. URL http://www.sciencedirect.com/science/article/pii/S0191814118300646.

Sérgio H. Faria, Ilka Weikusat, and Nobuhiko Azuma. The microstructure of polar ice. Part I: Highlights from ice core research. *Microdynamics of Ice*, 61: 2–20, April 2014. ISSN 0191-8141. doi: 10.1016/j.jsg.2013.09.010. URL https://www.sciencedirect.com/science/article/pii/S0191814113001740.

Fabie Gillet-Chaulet, Olivier Gagliardini, Jacques Meyssonnier, Maurine Montagnat, and Olivier Castelnau. A user-friendly anisotropic flow law for ice-sheet modeling. *Journal of Glaciology*, 51(172):3–14, 2005. ISSN 0022-1430. doi: 10.3189/172756505781829584. URL https://www.cambridge.org/core/journals/journal-of-glaciology/article/userfriendly-anisotropic-flo

J. W. Glen. Experiments on the Deformation of Ice. *Journal of Glaciology*, 2(12):111–114, 1952. ISSN 0022-1430. doi: 10.3189/S0022143000034067.

F. S. Graham, M. Morlighem, R. C. Warner, and A. Treverrow. Implementing an empirical scalar constitutive relation for ice with flow-induced polycrystalline anisotropy in large-scale ice sheet models. *The Cryosphere*, 12(3):1047–1067, 2018. doi: 10.5194/tc-12-1047-2018. URL https://www.the-cryosphere.net/12/1047/2018/.

Sigfús Jóhann Johnsen. GRIP Temperature Profile, January 2003. URL https://doi.org/10.1594/PANGAEA.89007.

B. Journaux, T. Chauve, M. Montagnat, A. Tommasi, F. Barou, D. Mainprice, and L. Gest. Recrystallization processes, microstructure and crystallographic preferred orientation evolution in polycrystalline ice during high-temperature simple shear. *The Cryosphere*, 13(5):1495–1511, May 2019. ISSN 1994-0424. doi: 10.5194/tc-13-1495-2019. URL `https://www.the-cryosphere.net/13/1495/2019/`.

M.-G. Llorens, A. Griera, P. D. Bons, I. Weikusat, D. J. Prior, E. Gomez-Rivas, T. de Riese, I. Jimenez-Munt, D. García-Castellanos, and R. A. Lebensohn. Can changes in deformation regimes be inferred from crystallographic preferred orientations in polar ice? *The Cryosphere*, 16(5):2009–2024, May 2022. ISSN 1994-0424. doi: 10.5194/tc-16-2009-2022. URL `https://tc.copernicus.org/articles/16/2009/2022/`. Publisher: Copernicus Publications.

Carlos Martín, G. Hilmar Gudmundsson, Hamish D. Pritchard, and Olivier Gagliardini. On the effects of anisotropic rheology on ice flow, internal structure, and the age-depth relationship at ice divides. *Journal of Geophysical Research: Earth Surface*, 114(F4), December 2009. ISSN 0148-0227. doi: 10.1029/2008JF001204. URL `https://doi.org/10.1029/2008JF001204`. Publisher: John Wiley & Sons, Ltd.

M. Montagnat, D. Buiron, L. Arnaud, A. Broquet, P. Schlitz, R. Jacob, and S. Kipfstuhl. Measurements and numerical simulation of fabric evolution along the Talos Dome ice core, Antarctica. *Earth and Planetary Science Letters*, 357-358:168–178, December 2012. ISSN 0012-821X. doi: 10.1016/j.epsl.2012.09.025. URL `https://www.sciencedirect.com/science/article/pii/S0012821X12005213`.

M. Montagnat, O. Castelnau, P.D. Bons, S.H. Faria, O. Gagliardini, F. Gillet-Chaulet, F. Grennerat, A. Griera, R.A. Lebensohn, H. Moulinec, J. Roessiger, and P. Suquet. Multiscale modeling of ice deformation behavior. *Microdynamics of Ice*, 61:78–108, April 2014. ISSN 0191-8141. doi: 10.1016/j.jsg.2013.05.002. URL `http://www.sciencedirect.com/science/article/pii/S0191814113000837`.

Erin C. Pettit, Throstur Thorsteinsson, H. Paul Jacobson, and Edwin D. Waddington. The role of crystal fabric in flow near an ice divide. *Journal of Glaciology*, 53(181):277–288, 2007. ISSN 0022-1430. doi: 10.3189/172756507782202766. URL `https://www.cambridge.org/core/product/67C81A182DA1D1F8F216E6C90558E948`. Edition: 2017/09/08 Publisher: Cambridge University Press.

Pierre Pimienta and Paul Duval. Mechanical behavior of anisotropic polar ice. *The Physical Basis of Ice Sheet Modelling*, 170:57–66, January 1987.

Luca Placidi, Ralf Greve, Hakime Seddik, and Sérgio H. Faria. Continuum-mechanical, Anisotropic Flow model for polar ice masses, based on an anisotropic Flow Enhancement factor. *Continuum Mechanics and Thermodynamics*, 22(3):221–237, March 2010. ISSN 1432-0959. doi: 10.1007/s00161-009-0126-0. URL `https://doi.org/10.1007/s00161-009-0126-0`.

C. Qi, D. J. Prior, L. Craw, S. Fan, M.-G. Llorens, A. Griera, M. Negrini, P. D. Bons, and D. L. Goldsby. Crystallographic preferred orientations of ice deformed in direct-shear experiments at low temperatures. *The Cryosphere*, 13(1):351–371, February 2019. ISSN 1994-0424. doi: 10.5194/tc-13-351-2019. URL `https://www.the-cryosphere.net/13/351/2019/`.

Nicholas M. Rathmann and David A. Lilien. Inferred basal friction and mass flux affected by crystal-orientation fabrics. *Journal of Glaciology*, pages 1–17, 2021. ISSN 1727-5652. doi: 10.1017/jog.2021.88.

Nicholas M. Rathmann and David A. Lilien. On the nonlinear viscosity of the orthotropic bulk rheology. *Journal of Glaciology*, 68(272):1243–1248, 2022. ISSN 0022-1430. doi: 10.1017/jog.2022.33. Edition: 2022/05/18 Publisher: Cambridge University Press.

Daniel H. Richards, Samuel S. Pegler, Sandra Piazolo, and Oliver G. Harlen. The evolution of ice fabrics: A continuum modelling approach validated against laboratory experiments. *Earth and Planetary Science Letters*, 556:116718, February 2021. ISSN 0012-821X. doi: 10.1016/j.epsl.2020.116718. URL `http://www.sciencedirect.com/science/article/pii/S0012821X20306622`.

Daniel H. Richards, Samuel S. Pegler, Sandra Piazolo, Nicolas Stoll, and Ilka Weikusat. Bridging the Gap Between Experimental and Natural Fabrics: Modeling Ice Stream Fabric Evolution and its Comparison With Ice-Core Data. *Journal of Geophysical Research: Solid Earth*, 128(11):e2023JB027245, November

2023. ISSN 2169-9313. doi: 10.1029/2023JB027245. URL https://doi.org/10.1029/2023JB027245. Publisher: John Wiley & Sons, Ltd.

---

## Author Response (AR2)

Dear Prof. Keegan,

Thank you for accepting the manuscript and your time through the review process. We have gone through the manuscript and supplementary material and made all these minor corrections as suggested. Please find the final manuscript, supplement and tracked changes included.

Yours sincerely,

Daniel Richards

---

## Author Response (AR3)

Dear Prof. Keegan,

Thank you for accepting the manuscript and your time through the review process. We have gone through the manuscript and supplementary material and made all the corrections highlighted in your report, which we include here for completeness:

- L16: do not remove 'is' from this first sentence
- L43-44: this is a sentence fragment that needs to be fixed – consider just removing 'While'
- L53: missing a word after 'by' here
- L115: add 'by' before the references or place them in parentheses; only the reference year needs to be put in parentheses for the Azuma and Goto-Azuma reference: '(1996)'
- L122: either 'grains' should be singular or 'grain's'
- L125: either remove 'to' or change it to 'the'
- L245: this sentence is unclear. Do you mean '...which act as two bounds that the true grain...'?
- L284: remove period after 'Equations'
- L314: check to make sure you aren't missing text here. If not, remove the hanging 'We'
- L375: 'infront' should be two words
- L379: should be 'implicitly'
- L384: need 'in' before 'the natural world'
- L402: 'of' before 'samples'
- L429: remove the extra 'surface' before 'create'
- L438: add 'that' or 'and' before 'extends'
- L473: add 'is' to the second sentence: 'The exception is...'
- L476: use in-text referencing here 'following Rathmann and Lilien (2021)'
- L477: use in-text referencing here: 'in Placidi and others (2010)'
- L498: 'this rheology' repeated twice here
- L536: 'grain's'
- L558: remove 'a' from Section 6.4 title
- L578: 'does' should be 'do'
- L594: 'where' should be 'were'
- L623: consider including the non-linear orthotropic acronym here '(NLO)', and use the acronym in subsequent references in this section (L626)
- L626: fix 'is it is'
- L632: remove the extra space before the comma; remove the period after 'Table'; add a comma after 'Table 2'
- L634: fix the section reference here
- L643: again consider including the non-linear orthotropic acronym here '(NLO)', and use the acronym in subsequent references in this section (Lines 651, 652, 654)
- L653: add 'is' after 'there'
- L657: define the VPSC acronym at its first use here
- L674: add 'that' after 'Now'
- L691: 'is' should be 'are'
- Supplement:
- L3: 'these' should be 'there'
- L20: 'the effect of the lattice rotation of stress,'
- L27: 'he' should be 'the'
- L33: add 'is' before 'no'
- L38-39: there are three uses of 'only' in this sentence, remove some for clarity

- Figure captions:
- Fig. S5: remove 'value' from 'value values'
- Fig. S7: remove 'value' from 'value values'
- Fig. S8: 'grain's'; remove 'value' from 'value values'
- Fig. S9: 'grain's'; remove 'value' from 'value values'
- Fig. S10: remove 'value' from 'value values'

We have also updated the code repository which reproduces the figures in the paper, so that it produces the updated figures for the revision (solely changes in the legend), as this was not done with the revision. Please find the final manuscript, supplement and tracked changes included.

Yours sincerely,

Daniel Richards